# The catalytic mechanism of the RNA methyltransferase METTL3

Ivan Corbeski[1], Pablo Andrés Vargas-Rosales[1], Rajiv Kumar Bedi[1], Jiahua Deng[2], Dylan Coelho[3], Emmanuelle Braud[3], Laura Iannazzo[3], Yaozong Li[1], Danzhi Huang[1], Mélanie Ethève-Quelquejeu[3], Qiang Cui[2,4,5], Amedeo Caflisch[1]*

[1]Department of Biochemistry, University of Zurich, Zurich, Switzerland; [2]Department of Chemistry, Boston University, Boston, United States; [3]Université Paris Cité, CNRS, Laboratoire de Chimie et Biochimie Pharmacologiques et Toxicologiques, Paris, France; [4]Department of Physics, Boston University, Boston, United States; [5]Department of Biomedical Engineering, Boston University, Boston, United States

*For correspondence:
caflisch@bioc.uzh.ch

**Abstract** The complex of methyltransferase-like proteins 3 and 14 (METTL3-14) is the major enzyme that deposits N⁶-methyladenosine (m⁶A) modifications on messenger RNA (mRNA) in humans. METTL3-14 plays key roles in various biological processes through its methyltransferase (MTase) activity. However, little is known about its substrate recognition and methyl transfer mechanism from its cofactor and methyl donor *S*-adenosylmethionine (SAM). Here, we study the MTase mechanism of METTL3-14 by a combined experimental and multiscale simulation approach using bisubstrate analogues (BAs), conjugates of a SAM-like moiety connected to the N⁶-atom of adenosine. Molecular dynamics simulations based on crystal structures of METTL3-14 with BAs suggest that the Y406 side chain of METTL3 is involved in the recruitment of adenosine and release of m⁶A. A crystal structure with a BA representing the transition state of methyl transfer shows a direct involvement of the METTL3 side chains E481 and K513 in adenosine binding which is supported by mutational analysis. Quantum mechanics/molecular mechanics (QM/MM) free energy calculations indicate that methyl transfer occurs without prior deprotonation of adenosine-N⁶. Furthermore, the QM/MM calculations provide further support for the role of electrostatic contributions of E481 and K513 to catalysis. The multidisciplinary approach used here sheds light on the (co) substrate binding mechanism, catalytic step, and (co)product release, and suggests that the latter step is rate-limiting for METTL3. The atomistic information on the substrate binding and methyl transfer reaction of METTL3 can be useful for understanding the mechanisms of other RNA MTases and for the design of transition state analogues as their inhibitors.

## eLife assessment

This **important** study combines experimental and computational data to address crucial aspects of RNA methylation by a vital RNA methyltransferase (MTase). The authors have provided **compelling**, strong evidence, utilizing well-established techniques, to elucidate aspects of the methyl transfer mechanism of methyltransferase-like protein 3 (METTL3), which is a part of the METTL3-14 complex. This work will be of broad interest to biochemists, biophysicists, and cell biologists alike.

## Introduction

### METTL3-14 is the main human mRNA m⁶A MTase

There are more than 170 RNA modifications forming the epitranscriptome (*Boccaletto et al., 2022*). N⁶-methyladenosine (m⁶A) is the most frequent internal modification of messenger RNA (mRNA)

**Figure 1.** Methyltransferase-like proteins 3 and 14 (METTL3-14) domain architecture and structure. (**A**) Domain architecture of METTL3 and METTL14. ZnF = zinc finger, NHM = N-terminal α-helical motif, CTM = C-terminal motif, RGG = arginine-glycine-glycine motif. (**B**) Crystal structure of the methyltransferase (MTase) domains of METTL3-14. Ribbon representations (left) are coloured as in panel (**A**). Surface renderings (right) are coloured according to the electrostatic potential. *S*-Adenosylmethionine (SAM) and the putative RNA binding site are indicated.

within the consensus sequence GGACU that is enriched near stop codons and in 3' untranslated regions (*Roundtree et al., 2017*; *Fu et al., 2014*; *Linder et al., 2015*). m⁶A affects most aspects of RNA regulation, i.e., alternative polyadenylation (*Ke et al., 2015*), splicing (*Ke et al., 2017*), nuclear export (*Lesbirel and Wilson, 2019*), stability (*Lee et al., 2020*), and translation initiation (*Fu et al., 2014*; *Kadumuri and Janga, 2018*). The complex of methyltransferase-like protein 3 (METTL3) and METTL14 (abbreviated as METTL3-14 in the following) is the main m⁶A-RNA methyltransferase (MTase) (*Liu et al., 2014*).

The METTL3-14 heterodimer is involved in a wide variety of diseases including type 2 diabetes (*De Jesus et al., 2019*), viral infections (*Dang et al., 2019*), and several types of cancer (*Chen et al., 2019b*). METTL3-mediated m⁶A deposition is directly involved in the development of acute myeloid leukaemia (AML) by promoting the translation of genes involved in cell growth, differentiation, and apoptosis (*Barbieri et al., 2017*; *Vu et al., 2017*). It has been demonstrated that inhibition of the METTL3 catalytic function is sufficient to induce apoptosis and differentiation in AML cells and in a mouse model of the disease but not in normal non-leukaemic haematopoietic cells (*Moroz-Omori et al., 2021*; *Yankova et al., 2021*). While it has been well established that METTL3-14 dysregulation is related to cancer development, the role of METTL3-14 varies in different cancer types, i.e., it can act as oncogene or tumour suppressor (*Zeng et al., 2020*). Despite growing knowledge of the diverse pathways involving METTL3-14, the mechanism of how m⁶A regulates gene expression remains poorly understood. Little is known about the recognition of specific RNA transcripts, the binding/release of the adenosine/m⁶A substrate/product, and the methyl transfer mechanism catalysed by METTL3. Furthermore, inhibiting the MTase function of METTL3-14 is a promising therapeutic strategy for several diseases (*Yankova et al., 2021*; *Dolbois et al., 2021*). Hence, understanding the mechanism of this complex would be helpful to develop new therapies.

METTL3-14 is the catalytic complex that transfers the methyl group from *S*-adenosylmethionine (SAM) to the substrate adenosine (*Figure 1*; *Wang et al., 2016b*; *Wang et al., 2016a*; *Śledź and Jinek, 2016*; *Yoshida et al., 2022*). METTL3 comprises a low-complexity region at the N-terminus, a zinc finger responsible for substrate binding, and the catalytic MTase domain at the C-terminus (*Figure 1A*; *Huang et al., 2019*). The METTL3 MTase domain has the catalytically active SAM binding site and adopts a Rossmann fold that is characteristic of Class I SAM-dependent MTases (*Figure 1B*). METTL14 has an MTase domain, too, however, with a redundant active site of hitherto unknown

function, and so-called RGG repeats at its C-terminus essential for RNA binding (*Yoshida et al., 2022*). METTL14 plays a structural role for complex stabilisation and RNA binding. It forms a positively charged groove at the interface with METTL3 which is predicted to be the RNA binding site.

## MD simulations for mechanistic studies of RNA MTases

Thus far, and to the best of our knowledge, no computational study of the conformational landscape or catalytic mechanism of eukaryotic RNA MTases has been reported. Molecular dynamics (MD) studies have mainly focused on protein MTases, and MTases from bacteria (*Sun et al., 2019*; *Sun et al., 2021*; *Singh et al., 2022*; *Singh et al., 2016*). In the latter, dynamic cross-correlation analysis, a technique usually applied for the study of allosteric processes (*Ichiye and Karplus, 1991*), showed that MTase conformational changes can influence the orientation of the substrate (*Singh et al., 2016*). In the former, Chen et al. explored the conformational landscape of SETD8, a histone MTase (*Chen et al., 2019a*). The study showed how slow conformational motions and conformational states of the MTase are relevant to catalysis.

Among RNA MTases, research has focused on viral enzymes. One example is the MD simulation analysis of the binding mechanism of S-adenosylhomocysteine (SAH) and $m^7$GTP to the Zika virus NS5 protein (*Chuang et al., 2018*). This enabled a detailed analysis of their interaction and understanding of the effects of MTase inhibitors as antiviral drugs. Another study focused on the mechanisms of the SARS-CoV-2 MTase nsp16 and its heterodimeric partner nsp10 which acts as a stimulator of SAM binding (*Sk et al., 2020*). The study provides a comprehensive understanding of the dynamic, thermodynamic, and allosteric processes of MTase complex formation and function.

## Bisubstrate analogues as structural tools to investigate the mechanisms of MTases

Only a few structures of RNA-bound MTases are currently available due to the intrinsic instability of RNA and RNA-enzyme complexes and resulting difficulties in obtaining their structures. Of the known $m^6$A RNA MTases, only METTL16, a SAM homeostasis factor, has been crystallised in complex with substrate RNA (*Doxtader et al., 2018*). The lack of more such structures results in poorly understood RNA recognition and methyl transfer mechanisms. In contrast, structures are available for many MTases in complex with either the cosubstrate SAM or the coproduct SAH, allowing for a well-understood cofactor binding within this protein family. Cofactor binding guided the initial design of bisubstrate analogues (BAs) as chemical tools to study the catalytic mechanisms of $m^6$A-MTases (*Figure 2*; *Oerum et al., 2019*).

BAs aim to mimic the transition state in which both the substrate nucleoside and the cosubstrate SAM are bound in the catalytic pocket of the enzyme while the methyl group is transferred from SAM to the adenosine $N^6$-atom of the substrate RNA during catalysis (*Figure 2A*; *Oerum et al., 2019*; *Atdjian et al., 2018*; *Atdjian et al., 2020*; *Meynier et al., 2022*; *Coelho et al., 2023*). They consist of a SAM analogue (5'N-SAM) covalently linked to the $N^6$ position of an adenosine of a ribonucleotide(-like) fragment (*Figure 2B*; *Atdjian et al., 2018*; *Schapira, 2016*). The only structural information on these molecules is their binding mode in the bacterial $m^6$A RNA MTase RlmJ (*Oerum et al., 2019*; *Meynier et al., 2022*). There, in the mononucleoside-containing compounds BA2 and BA4, the substrate adenosine is positioned in the presumed substrate binding pocket of RlmJ, and for the cofactor moiety, the methionine chain is bound like SAM. However, the structural studies resulted in a (co)substrate conformation that is not always biologically relevant, as the adenosine of the SAM analogue was rotated out of the canonical SAM binding pocket caused by π-stacking with the substrate adenine ring (*Oerum et al., 2019*). Therefore, the binding modes revealed therein are diverse and not always suitable for mechanistic studies. In a subsequent study, using a dinucleotide containing BA (GA* in *Figure 2B*), the SAM moiety had the correct orientation (*Meynier et al., 2022*). Furthermore, the $N^6$-atom of adenosine was positioned, through the alkyl chain of the linker, at 3 Å distance from the carbon corresponding to the Cε-atom of the methionine moiety in the cosubstrate SAM. Such a distance, in two non-linked moieties, would allow for an SN2 methyl transfer from SAM to the $N^6$-atom of adenosine (*O'Hagan and Schmidberger, 2010*). We therefore hypothesised that targeting human RNA MTases, in particular METTL3, with BAs might lead to structures that provide a suitable basis for understanding their MTase mechanism.

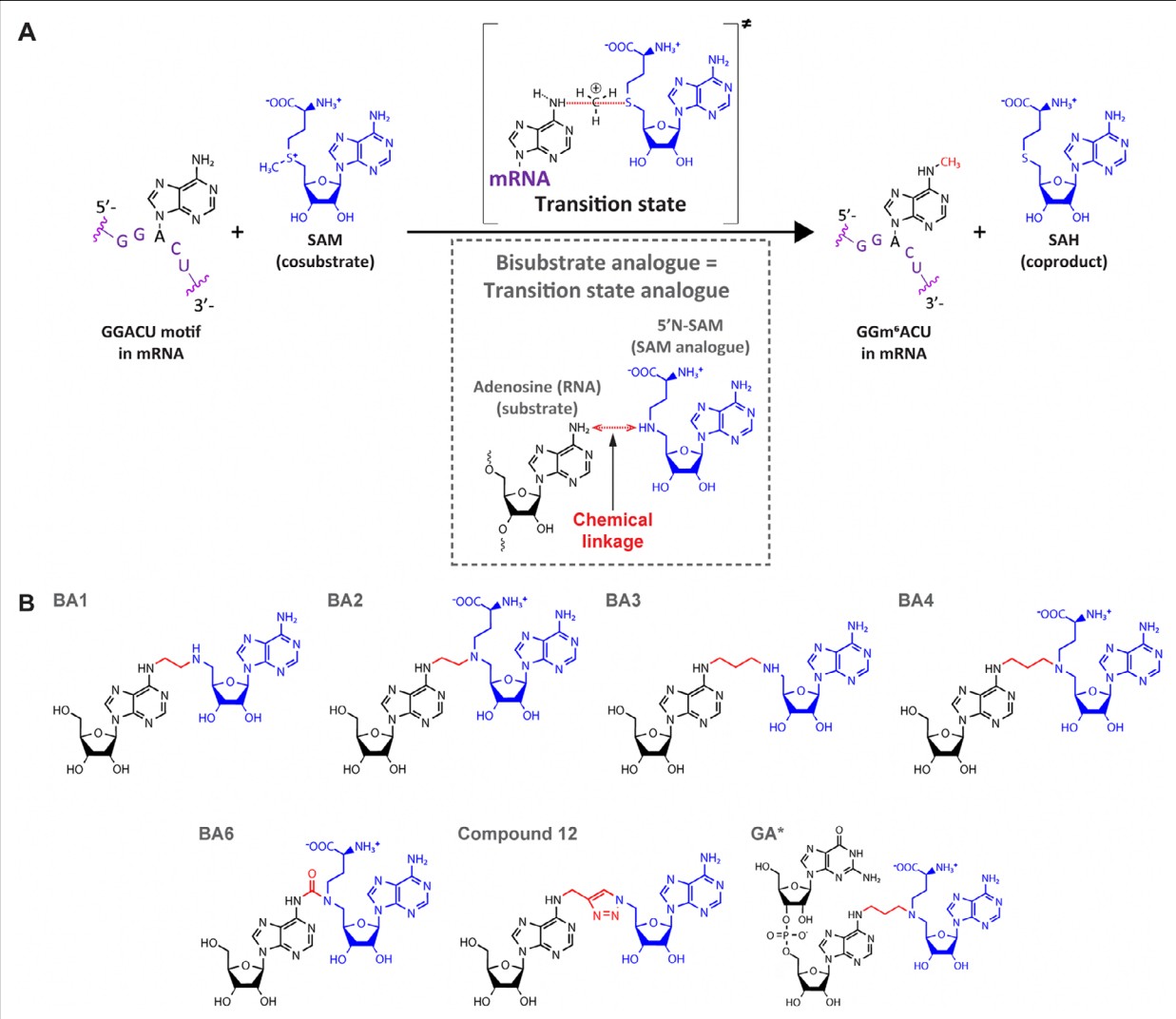

**Figure 2.** Bisubstrate analogues (BAs) as transition state mimics for methyltransferase-like protein 3 (METTL3). (**A**) METTL3-catalysed transfer of the methyl group of *S*-adenosylmethionine (SAM) to the N[6]-atom of A in a GGACU motif-containing messenger RNA (mRNA) and the production of N[6]-methyladenosine (m[6]A) and *S*-adenosylhomocysteine (SAH). The inset shows the design principle of BAs as transition state analogues. The point of linkage in the BA is indicated with a double arrow (red) between the N[6]-atom of adenosine and 5'N of the SAM analogue. (**B**) Chemical structures of the BAs used in this study. Substrate adenosine = black; SAM analogue = blue; linker = red. Compound names are as previously published: BA1/2/3/4/6, *Oerum et al., 2019*; Compound 12, *Atdjian et al., 2020*; GA*, *Meynier et al., 2022*.

The online version of this article includes the following figure supplement(s) for figure 2:

**Figure supplement 1.** Bisubstrate analogues show dose-dependent inhibitory effects in a time resolved-Förster resonance energy transfer (TR-FRET)-based enzymatic assay of methyltransferase-like proteins 3 and 14 (METTL3-14).

In the present study, we use a multidisciplinary approach to study the substrate binding, m[6]A methylation reaction, and release of m[6]A product in the human RNA MTase METTL3-14. To shed light on its catalytic mechanism, we combine crystal structures of METTL3-14-BA complexes with in vitro experiments, multiscale atomistic simulations, namely classical MD and quantum mechanics/molecular mechanics (QM/MM) free energy calculations. Crystal structures show the binding mode of the substrate adenosine in two different conformations, representing an encounter complex of RNA binding and the transition state of catalysis, respectively. These structures are validated through mutational analysis. Classical MD simulations are used to investigate the binding of the substrates SAM and adenosine and dissociation of the products SAH and m[6]A. QM/MM free energy calculations reveal the details of the methylation reaction. Taken together, we elucidate the reaction catalysed by METTL3-14

**Table 1.** Bisubstrate analogues (BAs) for methyltransferase-like proteins 3 and 14 (METTL3-14) characterised in this study.

| Bisubstrate analogue name[a] | IC$_{50}$ (µM)[b] ± SE | PDB ID[c] | Resolution[c] (Å) |
|---|---|---|---|
| BA1 | 346±66 | 8PW9 | 2.3 |
| BA2 | 9±1 | 8PW8 | 2.3 |
| BA3 | 21±2 | NA | NA |
| BA4 | 18±3 | 8PWA | 2.1 |
| BA6 | ≳500 | 8PWB | 2.5 |
| Compound 12 | ≳500 | NA | NA |
| GA* | 32±3 | NA | NA |

NA = not available.

[a] Compound names as previously published: BA1/2/3/4/6, **Oerum et al., 2019**; Compound 12, **Atdjian et al., 2020**; GA*, **Meynier et al., 2022**.

[b] IC$_{50}$=Half maximal inhibitory concentration from the enzymatic assay, SE=standard error from the fit; for comparison, IC$_{50}$ of S-adenosylhomocysteine (SAH) is 0.51µM, **Wiedmer et al., 2019**.

[c] Data for the METTL3-14-BA complex structures deposited to the Protein Data Bank (PDB).

at atomic level of detail. This knowledge will help in the further investigation of other MTases and the optimisation of chemical probes that target their function.

## Results and discussion

### Bisubstrate analogues bind in the METTL3 active site

We evaluated a series of bisubstrate analogues (BAs) as catalytic inhibitors and investigated the structural similarity between their binding mode in METTL3-14 and the putative RNA substrate and SAM cosubstrate during methyl transfer (see *Figure 2B*). First, we measured the inhibitory activity of the BAs on METTL3-14 enzymatic activity (*Table 1* and *Figure 2—figure supplement 1*). For this, we used our in-house developed reader-based enzymatic assay that quantifies the adenosine-N$^6$ methyl transfer based on homogeneous time-resolved fluorescence (HTRF) (*Wiedmer et al., 2019*). The assay detects m$^6$A using the natural m$^6$A-reader YTHDC1. The m$^6$A-RNA and -reader are fluorescently labelled such that their proximity during binding causes Förster resonance energy transfer (FRET). The BAs that inhibit METTL3 reduce the m$^6$A level, and thus decrease the FRET signal, in a dose-dependent manner. The low micromolar inhibitory activity of the most potent BAs motivated us to conduct a structural investigation on their binding mode to METTL3-14. We conducted crystallisation trials and obtained crystal structures of four of the BAs (BA1, BA2, BA4, and BA6) by soaking them into METTL3-14 crystals (see *Table 1* and *Supplementary file 1*).

All the crystallised BAs bind in the METTL3 active site (*Figure 3A*). BA1 and BA6, that are missing the methionine part of the SAM analogue or have a polar urea group in the linker, respectively, have the highest IC$_{50}$ values of the crystallised compounds in the enzymatic assay (see *Table 1*). Furthermore, in their crystal structures with METTL3, they reveal only a subset of the interactions of their SAM-like moiety compared to SAM (*Figure 3—figure supplement 1*). For BA2 and BA4, however, the interaction of their SAM-like moiety is the same as for SAM (*Figure 3B–F* and *Figure 3—figure supplement 2*). Both ligands bind with their SAM moiety in the active site lined with METTL3 hydrophobic residues and form polar contacts to conserved residues, namely D395, R536, H538, and N539 via the methionine part and D377, I378, F534, N549, and Q550 via the adenosine moiety of 5'N-SAM. These structures therefore validate their further analysis for the interpretation of the catalytic mechanism of METTL3.

### Adenosine has two distinct binding conformations in the METTL3 active site

In the METTL3-BA2 and -BA4 structures, the SAM moiety is superimposable with the METTL3-bound conformation of SAM, and adenosine is bound in the presumed catalytic site (*Figure 4A*). Adenosine,

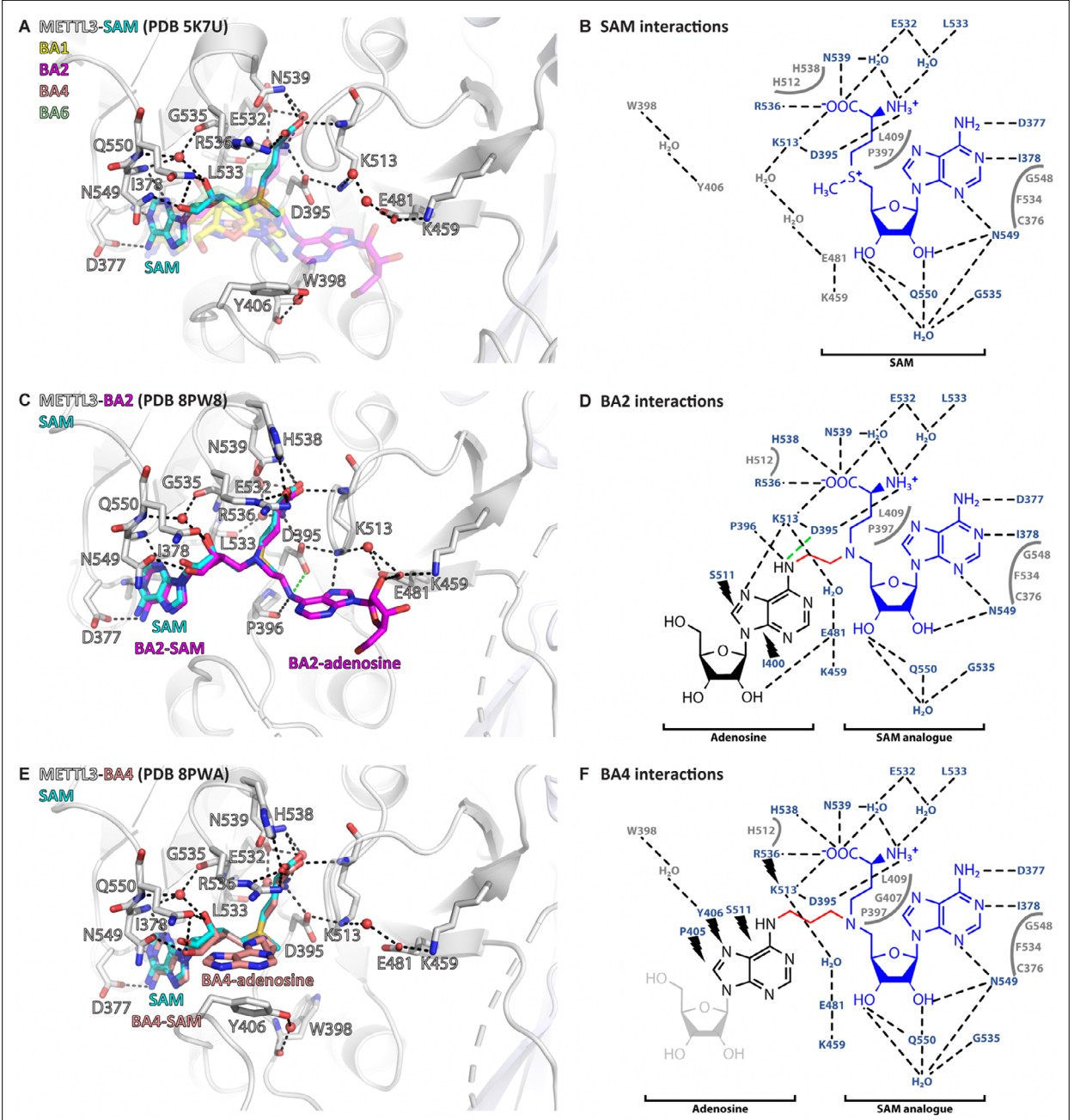

**Figure 3.** Crystal structures of methyltransferase-like proteins 3 and 14 (METTL3-14) show that bisubstrate analogues (BAs) bind in the METTL3 active site. (**A**) Superposition of the crystal structures of METTL3-14 bound to *S*-adenosylmethionine (SAM) and the four BAs. METTL3 backbone is shown as ribbon, side chains involved in polar interactions with SAM or intramolecularly are shown as sticks, waters as red spheres. SAM (cyan) is shown as sticks and indicated, and BAs are shown as transparent sticks (BA1=yellow, BA2=magenta, BA4=salmon, BA6=pale green). Black dashes indicate polar contacts in the crystal structure. (**B**) Outline of METTL3/SAM interactions from a LigPlot+ analysis (*Laskowski and Swindells, 2011*). Black dashed lines indicate polar contacts in the crystal structure, residues forming the binding pocket environment are shown in grey. (**C**) Structure of METTL3-BA2. Figure composition as in (**A**). BA2 is coloured magenta, its SAM and adenosine moieties are indicated. The green dashes indicate a hydrogen bond that does not form with BA2, but is likely to have favourable geometry to form between D395 and adenosine-N⁶ (i.e. the NH₂ group) of the natural RNA substrate. (**D**) Outline of METTL3/BA2 interactions from a LigPlot+ analysis, as in (**B**). The SAM analogue and adenosine parts of the BA are indicated. Black lightnings highlight residues in METTL3 involved in hydrophobic contacts with the adenosine moiety of the BA. (**E**) Structure of METTL3-BA4. Figure composition as in (**C**). BA4 is coloured salmon, its SAM and adenosine moieties are indicated. Note that BA4 is missing the ribose of the substrate adenosine moiety due to lack of electron density in the crystal structure probably due to flexibility of this group. (**F**) Outline of METTL3/BA4 interactions from a LigPlot+ analysis, as in (**D**). The missing ribose of the substrate adenosine moiety in the crystal structure is indicated with a lighter colour.

*Figure 3 continued on next page*

*Figure 3 continued*

The online version of this article includes the following figure supplement(s) for figure 3:

**Figure supplement 1.** Crystal structures of the complex of methyltransferase-like proteins 3 and 14 (METTL3-14) with the bisubstrate analogues BA1 and BA6 show divergent interactions with the *S*-adenosylmethionine (SAM) moiety.

**Figure supplement 2.** Crystal structures of the complex of methyltransferase-like proteins 3 and 14 (METTL3-14) with BA2 and BA4 show electron density supporting the conformations of the bisubstrate analogues (BAs) and their interactions with METTL3.

based on BA2 and BA4, is involved in an intricate network of interactions with side chains of METTL3 (*Figure 4B*).

The classical m⁶A MTase catalytic [395]DPPW[398] motif (D/N-PP-Y/F/W) of METTL3 in its flexible active site loop 1 (ASL1, METTL3 residues 395–410) participates to the binding of adenosine. D395 forms a salt bridge to SAM and its mutation to alanine was previously shown to abolish METTL3 activity and SAM binding, confirming its involvement in cosubstrate binding (*Wang et al., 2016b*; *Wang et al., 2016a*; *Śledź and Jinek, 2016*). The METTL3-BA2 structure reveals that there can also be a hydrogen bond formed between D395 and the N⁶-atom of adenosine in the case of the non-alkylated RNA substrate (*Figure 4B*). P396 in the [395]DPPW[398] motif, through its carbonyl group, forms a hydrogen bond to the N⁶-atom of adenosine. In addition, P397 from the [395]DPPW[398] motif makes hydrophobic contacts with SAM. The BA2 structure further shows that adenosine is stabilised by additional hydrogen bonds from its N⁷- and O2'-atoms to the METTL3 side chains of E481 and K513, respectively. While W398 from the [395]DPPW[398] motif is not directly involved in adenosine binding, its backbone carbonyl group forms a water-bridged hydrogen bond with the Y406 side chain in the BA4 structure. The latter is involved in hydrophobic interactions with adenosine in the BA4 structure whereas it remains flexible in the BA2-bound METTL3 structure (see *Figure 4A*).

Taken together, adenosine swaps conformation from solvent exposed in the METTL3-BA4 structure to buried in the METTL3-BA2 structure where it forms hydrogen bonds with residues in the active site loop 2 (ASL2, METTL3 residues 507–515). Each moiety of the BAs is involved in hydrophobic interactions with METTL3 residues, and both adenines (of SAM and adenosine) form hydrogen bonds to conserved residues in METTL3. Alanine mutation of D395, Y406, E481, and K513, which are involved in adenosine binding as seen in the BA2 and BA4 structures, almost completely abolishes the METTL3 catalytic activity (*Figure 4C*). Importantly, these residues are highly conserved in METTL3 (*Figure 4—figure supplement 1*). The loss of activity of the Y406, E481, and K513 mutants originates mainly from abolished binding capability to adenosine and not SAM, because they can still bind SAH, as seen from a thermal shift assay (TSA) (*Figure 4D*). Upon binding SAH, the thermal shift for the Y406, E481, and K513 mutants is similar as for the wild-type (WT) which suggests that their side chains are not involved in binding SAM/SAH but rather the RNA substrate. In contrast, mutation of D395 impairs SAH binding, which is consistent with the involvement of the D395 side chain in SAM binding (*Wang et al., 2016b*).

## MD simulations reveal BA2 as the stable adenosine binding pose

Because of the different conformations of METTL3 and adenosine in the BA2 and BA4 structures, we went on to characterise the enzyme dynamics in the presence of adenosine. We carried out multiple MD simulations of the METTL3-14 heterodimer in complex with (co)substrates and (co)products. Apo trajectories were generated in a previous study (*Bedi et al., 2023*). The MD simulations were started from the crystal structures of the complex with bisubstrate analogues BA2 and BA4 described above. Based on the position of each BA, the (co)substrates SAM and adenosine monophosphate (AMP) or the (co)products SAH and m⁶AMP were positioned in the protein, aligning them to their respective moiety of the BA. The intermolecular and intra-protein salt bridges and hydrogen bonds were monitored throughout the trajectories. The dissociation time for each of the substrates and products was analysed and modelled by fitting a single exponential to the fraction of bound ligands (*Table 2* using block averaging, and *Figure 4—figure supplement 2* considering all trajectories). The SAM and SAH cofactors remained bound in all but one of the sampled trajectories and thus fitting was not possible. The experimentally determined $k_{off}$ rates of SAM and SAH are $8\times10^{-4}$ s⁻¹ and $2\times10^{-4}$ s⁻¹, respectively, which corresponds to mean lifetimes of more than 20 min and 83 min, respectively (*Selberg et al., 2019*). Hence, we would not expect to observe dissociations of SAM or SAH in our 500 ns MD simulations.

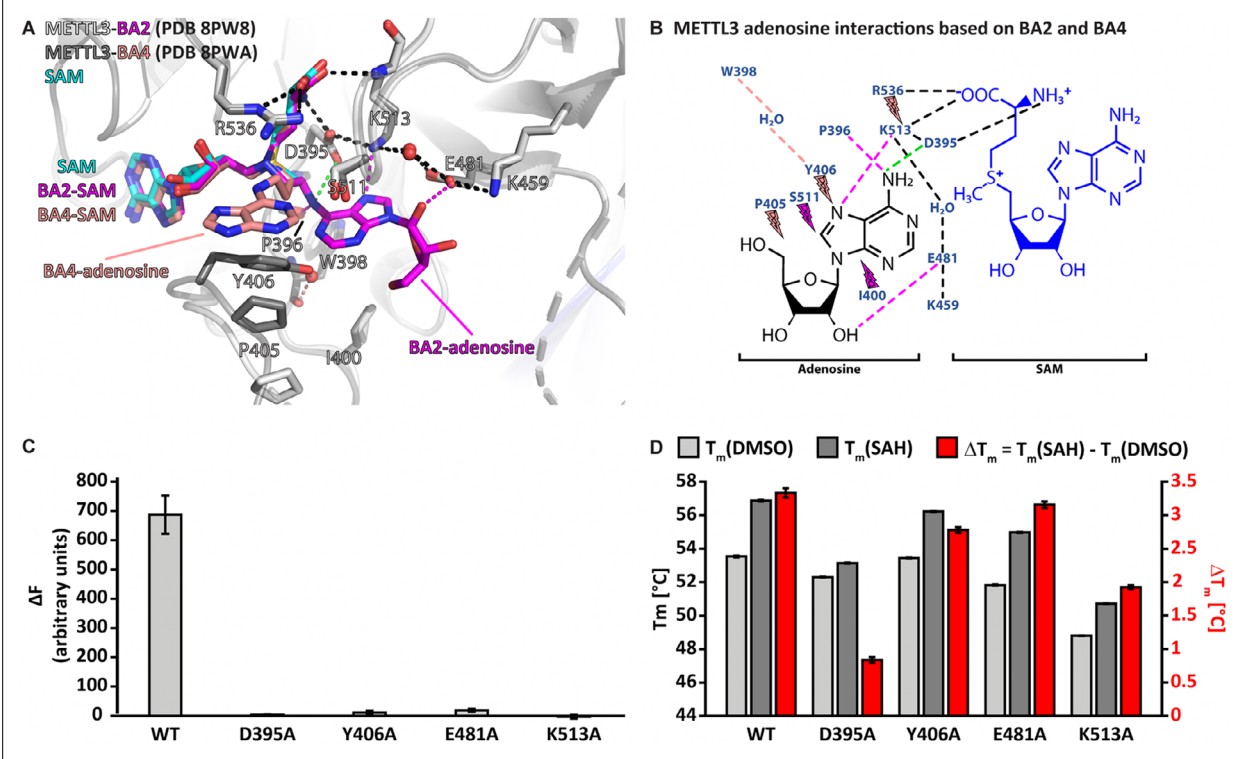

**Figure 4.** Crystal structures of methyltransferase-like proteins 3 and 14 (METTL3-14) with BA2 and BA4 reveal two distinct adenosine binding modes. (**A**) Superposition of the structures of *S*-adenosylmethionine (SAM) (cyan), BA2 (magenta), and BA4 (salmon) bound to METTL3. The ligands and their moieties are indicated. METTL3 in light/dark grey for BA2/BA4, backbone is shown as ribbon with side chains involved in the interactions with the adenosine moiety of the bisubstrate analogues (BAs) shown as sticks. Waters are shown as red spheres. Black dashes indicate polar contacts common to BA2/BA4. Magenta/salmon dashes indicate hydrogen bonds unique to BA2/BA4. The green dashes indicate a hydrogen bond that does not form with BA2, but is likely to have favourable geometry to form between D395 and adenosine-$N^6$ (i.e. the $NH_2$ group) of the natural RNA substrate. Note that there is no electron density for the side chain of Y406 in the complex with BA2 and for the ribose of the adenosine moiety in the complex with BA4 which is most likely due to flexibility of these groups. (**B**) Ligplot+ analysis showing key interactions between METTL3, adenosine, and SAM based on the BA2 and BA4 structures. Dashed lines indicate polar contacts as in (**A**). Small magenta/salmon lightnings highlight residues in METTL3 involved in hydrophobic contacts with the adenosine moiety in the BA2/BA4 conformation. (**C**) Mutational analysis of the enzymatic activity of METTL3 active site residues involved in adenosine binding. The error bars represent standard deviation from triplicate measurements. (**D**) The melting temperature ($T_m$) and its shift ($\Delta T_m$, in red) for METTL3 wild-type (WT) and mutants with DMSO as control (light grey bars) or in the presence of *S*-adenosylhomocysteine (SAH) (dark grey bars) measured using differential scanning fluorimetry. The error bars represent standard deviation from triplicate measurements.

The online version of this article includes the following figure supplement(s) for figure 4:

**Figure supplement 1.** Methyltransferase-like protein 3 (METTL3) residues that interact with the adenosine part of the bisubstrate analogues are highly conserved.

**Figure supplement 2.** Exponential fitting of adenosine monophosphate (AMP) and m⁶AMP dissociation.

**Figure supplement 3.** Geometric annotation for trajectories started with substrates in the BA4 crystal structure conformation.

**Figure supplement 4.** Geometric annotation for trajectories started with products in the BA4 crystal structure conformation.

**Figure supplement 5.** Geometric annotation for trajectories started with substrates in the BA2 crystal structure conformation.

**Figure supplement 6.** Geometric annotation for trajectories started with products in the BA2 crystal structure conformation.

**Figure supplement 7.** Structural stability and conformational transitions of adenosine monophosphate (AMP) in the molecular dynamics (MD) simulations.

**Figure supplement 8.** Geometric annotation of apo trajectories.

Simulations that were started with AMP or m⁶AMP in the conformation of BA4 showed immediate dissociation of these ligands (see *Table 2* and *Figure 4—figure supplements 3 and 4*). This could mean that this conformation represents a short-lived intermediate during substrate binding or product release. Possibly, it represents an encounter complex between the METTL3-SAM holo complex and

**Table 2.** Kinetic parameters of ligand dissociation.

Mean lifetime ($\tau$) in ns of the analysed (co)substrates or (co)products as calculated from fitting of a single exponential (A) or an exponential with a multiplicative factor (in parentheses) (B).

| Initial structure | Fit | Substrates | | Products | |
|---|---|---|---|---|---|
| | | SAM | AMP | SAH | m⁶AMP |
| BA2 | A | >>500 | 426±196 | >>500 | 12.87±0.02 |
| | B | | 579±386 (0.9±0.1) | | 12.70±4.95 (1.1±0.1) |
| BA4 | A | >>500 | 0 | >>500 | 0 |

RNA substrate. This could be promoted through electrostatic steering of the negatively charged RNA backbone by the positively charged sulphonium ion of SAM.

The simulations started from the BA2 conformation show that the AMP and m⁶AMP ligands dissociated in several of the analysed MD simulation runs (*Figure 4—figure supplements 5 and 6*). Binding of AMP is more stable, while the methylated product dissociates more quickly (see *Table 2*). This difference originates, at least in part, from the long-range monopole-monopole electrostatic interaction between the positively charged SAM and the negatively charged AMP. After the methyl transfer reaction, SAH is neutral which does not favour interaction with m⁶AMP resulting in rapid dissociation of the latter.

One caveat is that we simulate only a mononucleotide, resulting in much faster dissociation of both the substrate AMP and product m⁶AMP than would be expected for the longer, canonical RNA substrate/product. To the best of our knowledge, there are no published affinities of AMP and m⁶AMP for METTL3-14 binding. Based on the mean lifetimes determined from the MD simulations, and assuming diffusion-limited association rates of $10^9$ M$^{-1}$s$^{-1}$, the dissociation constants of AMP and m⁶AMP are ~2 and ~78 mM, respectively. In contrast to these low affinities, the long GGACU motif containing mRNA can interact with the binding groove at the METTL3-14 interface with dissociation constants in the nanomolar range, resulting in a more stable complex (*Qi et al., 2022*). Nevertheless, our model is useful as it emphasises the differences in binding affinity due to the different electrostatic interactions between the SAM-AMP and SAH-m⁶AMP pairs. Besides the structural stability of the BA2-like pose of AMP, the MD simulations also suggest that AMP can transiently populate a binding mode similar to BA4 (*Figure 4—figure supplement 7*). This further supports that BA4 could represent an intermediate binding conformation of adenosine.

## A polar interaction network stabilises the BA2 conformation of adenosine in METTL3

The crystal structure of the complex with BA2 reveals a string of ionic interactions which involves charged side chains of METTL3 and the amino and carboxyl groups of the SAM analogue. The string consists of seven charged groups (four of which are positive): R536⁺ – SAM COO⁻ – SAM NH₃⁺ – D395⁻ – K513⁺ – E481⁻ – K459⁺ (see *Figure 4A and B*). We decided to monitor these salt bridges and the following monopole-dipole interactions in the MD simulations: D395⁻ – AMP-N⁶, K513⁺ – AMP-N⁷, and E481⁻ – AMP-2'OH (see *Figure 4—figure supplements 3–6*). In the BA2 conformation, the polar interactions between AMP and the charged side chains D395 and K513 are stronger than the interaction between E481 and the 2′ hydroxyl group of the ribose (see *Figure 4—figure supplement 5*, black traces). For SAM, the interaction between its positively charged amine and D395 is more stable than the one between its negatively charged carboxy with R536 which can adopt different conformations (see *Figure 4—figure supplement 5*, blue traces). The intramolecular salt bridges are also stable throughout the sampling (see *Figure 4—figure supplement 5*, grey traces). The salt bridge D395⁻ – K513⁺ seems to be the most stable, while E481⁻ – K459⁺ seems to change frequently between direct and water-separated contacts. In METTL3 apo trajectories, the D395⁻ – K513⁺ salt bridge is not present in the initial structure but is formed during the course of the simulation (*Figure 4—figure*

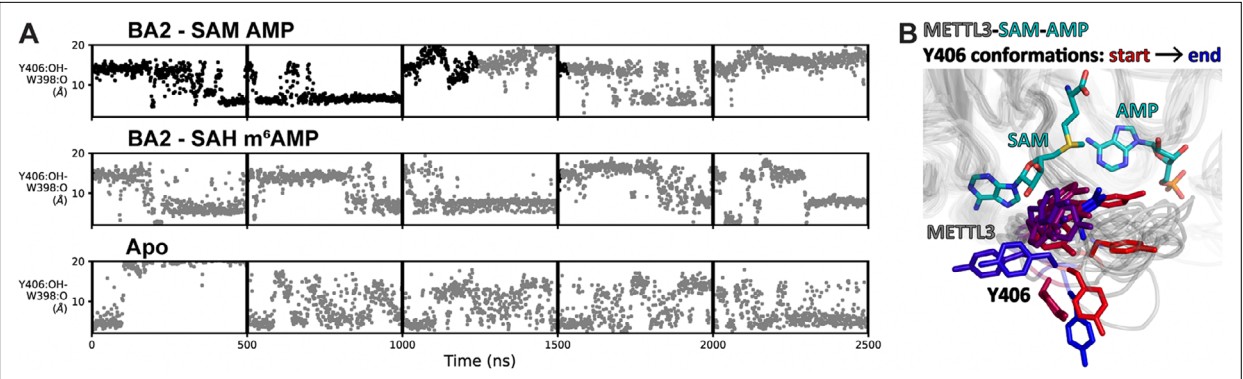

**Figure 5.** Flexibility of Y406 is evidenced by molecular dynamics (MD) simulations. (**A**) Distance time series of five 500 ns MD trajectories started from the BA2 conformation with substrates (top), products (middle), or apo methyltransferase-like proteins 3 and 14 (METTL3-14) (bottom). The distance between the Y406 side chain and the backbone O of W398 (grey trace) reports on the orientation of Y406 and the flexibility of the loop. The data points are coloured black if adenosine monophosphate (AMP)/m⁶AMP is bound, and grey if not. Bound AMP is defined by a distance of less than 6 Å between N⁶ of adenosine and Cγ of D395. (**B**) The conformations of the flexible METTL3 backbone (ribbon) and Y406 side chain (sticks) are shown at different timepoints between 0 and 300 ns of the simulation with S-adenosylmethionine (SAM) and AMP, the Y406 side chain is coloured from red/start to blue/end. SAM and AMP are shown as sticks at the start of the simulation.

The online version of this article includes the following video for figure 5:

**Figure 5—video 1.** Stable/flexible binding of S-adenosylmethionine (SAM)/adenosine monophosphate (AMP) and flexibility of ASL1/Y406 is evidenced by molecular dynamics (MD) simulations.

https://elifesciences.org/articles/92537/figures#fig5video1

supplement 8). The bond between E481 and K513 is stable but separated by water when SAM and AMP are bound. This contrasts with the METTL3 apo simulations, where this bond forms transiently. The simulations with the products reveal similar interactions except for the faster dissociation of m⁶AMP (see *Table 2* and *Figure 4—figure supplement 6*). Another difference is the weaker interaction between SAH and D395 which fluctuates more than with SAM. The intramolecular interactions are also observed for the simulations initiated from the BA4 conformation of adenosine (see *Figure 4—figure supplements 3 and 4*). These MD results validate the mechanistic interpretation of the BA-bound METTL3-14 crystal structures, and give a dynamic view of the behaviour of the complex before (substrate bound) and after (product bound) the methyl transfer reaction.

## The flexibility of METTL3 Y406 supports the recruitment of adenosine

In structures of METTL3-14 in the apo state or bound to SAH or SAM, METTL3 residue Y406 is found in different conformations (*Wang et al., 2016b*; *Wang et al., 2016a*; *Śledź and Jinek, 2016*). One study suggested that Y406 makes a hydrogen bond with S511 in ASL2 and thereby caps the SAH coproduct (*Śledź and Jinek, 2016*). However, another study suggested that Y406 might be important for the interaction with nucleotide bases (*Wang et al., 2016a*). The different conformations of Y406 seen in our crystal structures with BA2 and BA4 support the latter and suggest an involvement of the Y406 side chain in RNA nucleotide binding, probably as a first step of RNA recognition (see *Figure 4A*). Indeed, mutation of Y406 to alanine (in this and a previous study) or cysteine (in a previous study) abolishes MTase activity (see *Figure 4C*; *Wang et al., 2016a*; *Śledź and Jinek, 2016*).

The flexibility of Y406 in the MD simulations was analysed by monitoring the distance between its hydroxyl oxygen atom and the backbone carbonyl oxygen of METTL3 residue W398. There is a water-bridged polar interaction between these two oxygen atoms in the BA4 crystal structure (see *Figure 3E*). In the MD simulations, Y406 transitions multiple times (on the μs time scale) from orientations far away from the W398 carbonyl oxygen (distance of ~15 Å) to shorter distances of ~6 Å (*Figure 5A*). This corresponds to a transition from the extended ASL1, with Y406 pointing outside the pocket, to a conformation compatible with the water-bridged hydrogen bond observed in the crystal structure with BA4. *Figure 5B* shows a superposition of frames of a BA2 substrate MD trajectory illustrating this behaviour. We observe that the adenine ring system of AMP can be involved in a π-π interaction with Y406 for several ns before exiting the pocket, and is then captured again later

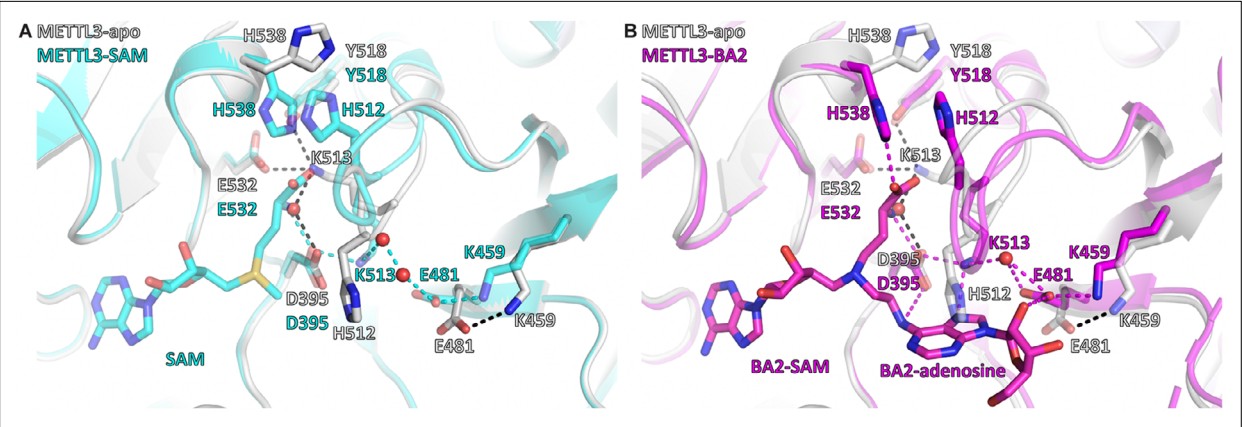

**Figure 6.** *S*-Adenosylmethionine (SAM) binding primes the methyltransferase-like protein 3 (METTL3) active site for adenosine binding. (**A**) Structural overlay of the METTL3 apo (grey) and SAM-bound holo state (cyan). Black/cyan dashes indicate intramolecular polar contacts in apo/holo METTL3. Residues are shown as sticks and labelled. SAM and METTL3 residues in the holo state are shown as transparent sticks. Waters are shown as red spheres. (**B**) Structural overlay of the METTL3 apo (grey) and BA2-bound state (magenta). Black dashes indicate intramolecular polar contacts in apo METTL3, magenta dashes indicate polar contacts in BA2-bound METTL3. Residues are shown as sticks and labelled. BA2 is shown in magenta as sticks, its SAM and adenosine moieties are indicated. METTL3 residues in the apo state are shown as transparent sticks. Waters are shown as red spheres.

The online version of this article includes the following figure supplement(s) for figure 6:

**Figure supplement 1.** Flexibility of the side chains of H512 and H538.

by the Y406 side chain, but is not brought back into the pocket (*Figure 5—video 1*). The full binding mechanism thus probably requires the rest of the substrate RNA, though the role of Y406 emerges already from the present simulations with the mononucleotide. The different conformations of Y406 reflect different steps in the binding and catalysis reaction. In the METTL3 SAM-bound state, the Y406 side chain is flexible. To bind RNA effectively, Y406 needs to stabilise through the water-bridged polar interaction with the backbone of W398. The substrate adenosine can then bind through selection of this conformation. Additionally, Y406 plays a role in positioning the adenosine substrate at the catalytic site.

## SAM binding primes the METTL3 active site for adenosine recognition

Given the high concentration of SAM in the cell (60–160 µM in the rat liver), the cosubstrate SAM is expected to bind before the RNA substrate (*Finkelstein and Martin, 1984*). SAM binding results in large conformational changes of several side chains in the METTL3 active site (*Figure 6*). In the apo state, the K513 side chain points away from the putative RNA binding site and is involved in intramolecular polar contacts with the side chains of Y518, E532, and, via a water molecule, D395, that probably help to stabilise the apo protein (*Figure 6A*). SAM binding disrupts the water-mediated hydrogen bond between K513 and D395. This leads to a conformational change in the K513 side chain which then points in the direction of the RNA binding site where it can form a direct salt bridge with D395. In that conformation, the K513 side chain can also readily form a hydrogen bond to the adenosine-N[7] of an RNA substrate as seen in the BA2 structure (*Figure 6B*). Furthermore, the side chain of H512, which is part of the SAM binding pocket environment, also undergoes a conformational change upon SAM binding (see *Figure 6A*). In the apo state, the H512 side chain points inwards and thus blocks the adenosine binding site. However, once SAM is bound, the H512 side chain is attracted to form a π-π interaction with H538 whose side chain rotates from an outward pointing conformation in the apo state to an inward pointing conformation in which it interacts with SAM in the holo state. This conformational change of the H512 side chain makes space for adenosine to bind as seen in the BA2 structure (see *Figure 6B*). MD simulations show that the side chains of both H512 and K513 are flexible in the apo state, but undergo stabilisation upon SAM and adenosine binding (*Figure 4—figure supplements 5 and 8* and *Figure 6—figure supplement 1*). Together, the conformational switches of the H512 and K513 side chains upon SAM binding can be seen as priming METTL3 for adenosine binding.

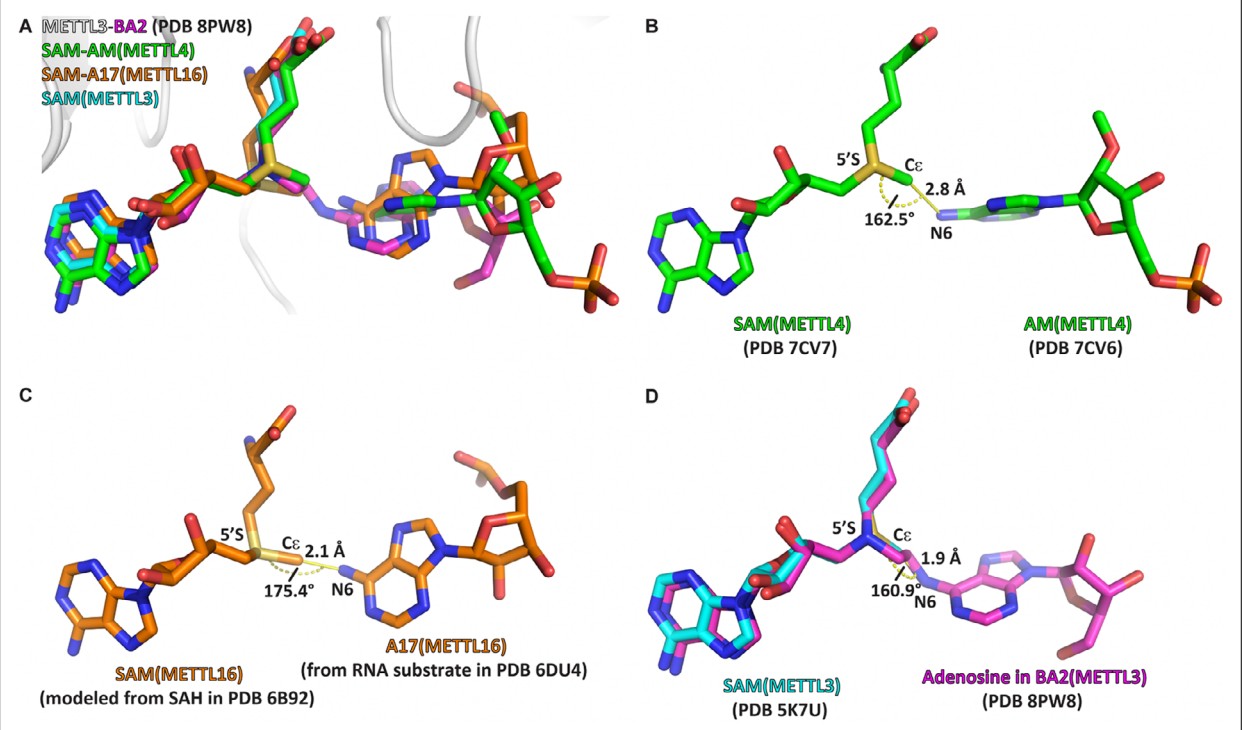

**Figure 7.** The crystal structure of the complex of methyltransferase-like proteins 3 and 14 (METTL3-14) with the bisubstrate analogue BA2 represents the transition state of catalysis. (**A**) Overlap of the crystal structure of the complex of METTL3-14 (grey) with BA2 (magenta) to the complex with *S*-adenosylmethionine (SAM) (cyan) and the substrate-cosubstrate pairs of METTL4 (green) and METTL16 (orange). The overlap was generated by aligning SAM from each (co)substrate pair to the SAM of METTL3. (**B–D**) Measurements of distances and angles between the adenosine-N⁶ and SAM-CH₃ groups in the respective (co)substrate pairs shown in (**A**). (**B**) The METTL4 (co)substrate pair was generated by aligning the structure of METTL4-AM to METTL4-SAM. (**C**) The METTL16 (co)substrate pair was generated by aligning the structure of METTL16-MAT2A 3'UTR hairpin 1 to METTL16-*S*-adenosylhomocysteine (SAH). SAM was then generated from SAH using the Chem3D software. (**D**) The METTL3 overlay was generated by aligning the structure of METTL3-BA2 to METTL3-SAM. The distance was measured between the N⁶ of BA2 and Cε of SAM, the angle was measured between the N⁶ of BA2, Cε of SAM, and 5'S of SAM.

## BA2 represents a transition state analogue of the METTL3 catalysed methyl transfer reaction

We compared the structure of METTL3-BA2 with the structures of RNA MTases METTL4 and METTL16 bound to their substrates (*Figure 7*). When the SAM moiety of BA2 is superimposed with SAM bound to METTL4 and METTL16, the adenosine moiety of BA2 is situated in a very similar position as the substrates of the other MTases (*Figure 7A*). In METTL4 and METTL16, the N⁶-atom of their substrate adenosine (analogue) is positioned at a distance of 2.8 and 2.1 Å from the methyl group of SAM, respectively (*Figure 7B and C*). The angle formed between SAM-5'S – Cε – adenosine-N⁶ in the substrate-cosubstrate pairs of METTL4 and METTL16 is 162.5° and 175.4°, respectively. In BA2, the adenosine-N⁶ is situated, through the alkyl chain of the linker, at 1.9 Å away from the carbon corresponding to the Cε-atom of the methionine moiety in the SAM cofactor (*Figure 7D*). The 1.9 Å distance in the crystal structure with BA2 is similar to the corresponding distance in METTL4 and METTL16, which, in two non-linked moieties, would allow for an SN2 transfer of the methyl group of a METTL3-bound SAM to the N⁶-atom of an adenosine substrate. Strikingly, the angle formed by the SAM-5'S – Cε – adenosine-N⁶ in METTL3 is 160.9° which is similar to the other MTases, and very close to the optimal 180° for an SN2 reaction in which the adenosine-N⁶ attacks the SAM-Cε and SAH becomes the leaving group. Hence, the adenosine of BA2 is in a suitable orientation for methyl transfer and BA2 represents a transition state mimic for METTL3. This is useful atomistic information for setting up QM/MM free energy calculations to study the catalytic reaction (see below).

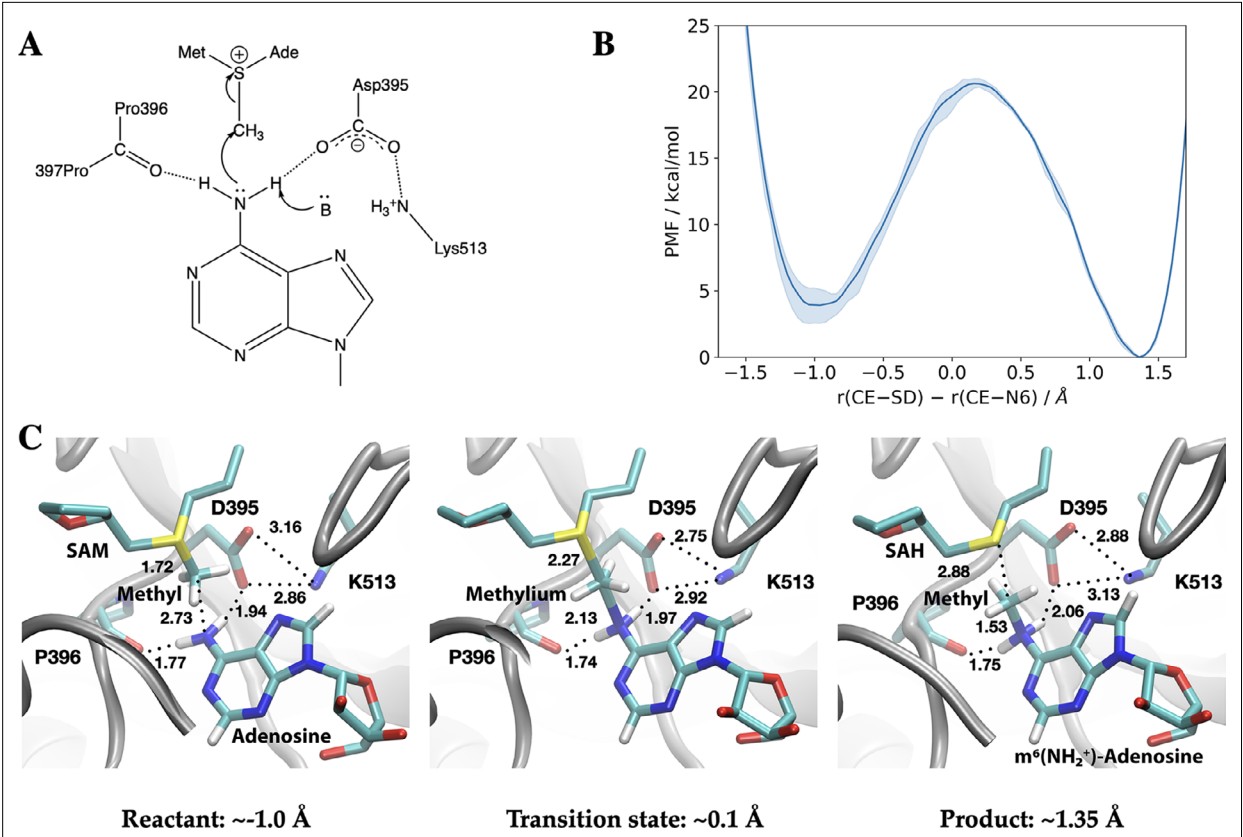

**Figure 8.** Methyl transfer catalysed by methyltransferase-like protein 3 (METTL3) without prior deprotonation of adenosine is energetically favourable based on third-order density functional tight binding (DFTB3)/MM simulations. (**A**) Proposed mechanism of the methyl transfer reaction catalysed by METTL3. (**B**) Potential of mean force (PMF) along the antisymmetric stretch coordinate that describes the transfer of the methyl group (carbon atom indicated as CE) between the *S*-adenosylmethionine (SAM) sulphur atom (indicated as SD) and the N$^6$-atom in adenosine (indicated as N6) computed using multiple walker metadynamics simulations. The solid curve represents the PMF averaged over three independent runs, and the shaded area represents the standard error of the mean of the three replicas. (**C**) Snapshots of the active site for the reactant (left panel), transition state (middle panel), and product windows (right panel). Key distances (in Å) involving the reactive groups and the nearby ion pair (D395-K513) and P396 backbone carbonyl are shown. METTL3 backbone is shown in grey ribbon representation with side chains shown as sticks and labelled. SAM/adenosine and *S*-adenosylhomocysteine (SAH)/m$^6$(NH$_2$$^+$)-adenosine are shown as sticks and labelled in the reactant and product window, respectively, together with the transferred methyl group. The methylium group (CH$_3$$^+$) is indicated in the transition state window.

The online version of this article includes the following figure supplement(s) for figure 8:

**Figure supplement 1.** Potentials of mean force computed with different numbers of deposited Gaussians during the metadynamics simulation are compared to illustrate the convergence behaviour of the simulation.

**Figure supplement 2.** The time series of the collective variable (CV) sampled by the 24 independent walkers during one set of metadynamics simulations.

## The METTL3 catalytic pocket supports direct methyl transfer without prior deprotonation

We carried out hybrid QM/MM (*Warshel and Levitt, 1976*; *Field et al., 1990*; *Senn and Thiel, 2009*; *Gao, 1995*; *Brunk and Rothlisberger, 2015*; *Chung et al., 2015*; *Lu et al., 2016*; *Hu and Yang, 2008*) free energy simulations to establish the catalytic mechanism of RNA methylation by the METTL3-14 complex (*Śledź and Jinek, 2016*; *Wu et al., 2017*; *Oerum et al., 2021*). The crystal structure with the bisubstrate analogue BA2 (see *Figure 3C*) was used as the starting point for the QM/MM simulations (*Figure 8*).

In the simplest mechanism (*Figure 8A*), the methyl cation in the SAM cofactor is transferred directly to the N$^6$ position of the adenosine substrate, prior to the deprotonation of the adenosine N$^6$H$_2$ group, which has a very high p$K_a$ of ~17 (*Lippert, 2005*). Indeed, third-order density functional tight binding (DFTB3)/MM free energy simulations show that this mechanism is energetically favourable (by

**Table 3.** Reaction energetics (in kcal/mol) computed for model methyl transfer reactions that involve *S*-adenosylmethionine (SAM) and adenosine using different levels of theory*.

| Reaction[†] | Different computational models | | | | | | |
| | B3LYP-D3 | ωB97XD | B3LYP-D3 | ωB97XD | DFTB3/3OB | B3LYP-D3 | ωB97XD |
| | aug-cc-pVDZ | | aug-cc-pVTZ | | | CPCM[‡], aug-cc-pVTZ | |
| SAM+Ade → SAH+Ade-$CH_3$ | 5.3 | 5.0 | 7.3 | 6.9 | 2.5 | 10.2 | 9.6 |
| SAM+dp-Ade → SAH+dp-Ade-$CH_3$ | −129.1 | −130.6 | −128.1 | −129.8 | −140.5 | −37.7 | −38.9 |
| Difference | −134.5 | −135.6 | −135.4 | −136.8 | −143.0 | −47.9 | −48.5 |

*Different density functional theory methods (B3LYP-D3, ωB97XD) and basis set combinations (aug-cc-pVDZ and aug-cc-pVTZ), along with a semi-empirical quantum mechanical method, third-order density functional tight binding (DFTB3)/3OB, are used to evaluate the intrinsic energetics for model methyl transfer reactions. The comparison also helps validate the more approximate DFTB3/3OB method, which is used in quantum mechanics/molecular mechanics (QM/MM) free energy simulations.

[†]dp-Ade indicates a deprotonated adenosine at the $N^6$ position. SAM and *S*-adenosylhomocysteine (SAH) are modelled by replacing the adenosine and amino moieties by ethyl groups.

[‡]A dielectric constant of 78.4 is used for the conductor-like polarisable continuum model (CPCM) to estimate the reaction energetics in solution. All other results are based on gas-phase calculations.

about ~4 kcal/mol) with a barrier of 15–16 kcal/mol (*Figure 8B* and *Figure 8—figure supplements 1 and 2*). The catalytic turnover of METTL3-14, as measured by an enzymatic RNA methylation assay, is 0.2–0.6 $min^{-1}$ at ambient temperature which implies a barrier of ~20 kcal/mol (*Buker et al., 2020*; *Garcia-Viloca et al., 2004*; *Glowacki et al., 2012*). Hence, the methyl transfer is not the rate-limiting step. Taken together, the QM/MM and MD simulations suggest that the dissociation of the coproduct SAH and product RNA is likely the rate-limiting step.

Compared to the model reaction in solution computed using a continuum solvation model (*Table 3*), the reaction in the enzyme is substantially more exoergic, suggesting that the enzyme environment stabilises the product of the methyl transfer reaction. Inspection of the active site structure based on DFTB3/MM simulations suggests that such stabilisation primarily comes from the hydrogen-bonding interactions between the adenosine-$N^6$ group and nearby polar groups, in particular the side chain of D395 and the backbone carbonyl of P396 (*Figure 8C*).

This catalytic mechanism is similar to $N^6$-adenine DNA methyl transferase M·TaqI, in which the adenosine-$N^6$ group is hydrogen-bonded to an Asn side chain and backbone carbonyl of a Pro in the protein (*Goedecke et al., 2001*). In the absence of any catalytic base, it was proposed that the methyl transfer occurs first, leading to an adenine-$m^6NH_2^+$ group well stabilised by hydrogen-bonding interactions with the Asn side chain and Pro backbone carbonyl. The mechanism was supported by QM/MM free energy simulations with a barrier height of ~20 kcal/mol (*Aranda et al., 2014*). On the other hand, the same QM/MM study suggested that when the active site Asn was replaced by an Asp, a mechanism in which deprotonation of the adenine-$N^6$ group by the Asp preceded the methyl transfer from SAM also had a comparable free energy barrier. Due to the involvement of the proton transfer, the corresponding transition state exhibited rather different charge distributions from that in the WT M·TaqI. DNA $N^6$-methyltransferases in the α/β groups feature an Asp in the active site, while those in the γ group have an Asn at the equivalent position (*Malone et al., 1995*). Hence, the QM/MM computational results suggest that transition states with distinct charge distributions are involved in different groups of enzymes (i.e. α/β vs. γ), giving rise to the opportunity of designing transition state analogues as inhibitors unique to specific classes of methyl transferases (*Aranda et al., 2014*).

This raises the question whether methyl transfer in METTL3-14 may also occur following deprotonation of the adenosine-$N^6H_2$. The structural features of the METTL3-14 active site do not support this mechanism. The adenosine-$N^6$ position has a very high $pK_a$ of ~17, and thus its deprotonation requires a particularly strong base, which is absent in the active site of METTL3-14. For example, while there is a carboxylate nearby (D395), it forms a salt bridge with K513, and therefore is expected to feature a too low $pK_a$ value to deprotonate the adenosine-$N^6H_2$. These considerations are congruent with the observation that the computed DFTB3/MM free energy profile without adenosine-$N^6$ deprotonation is possible within the reported experimental kinetics (*Buker et al., 2020*; *Woodcock et al., 2019*; *Xiao et al., 2022*).

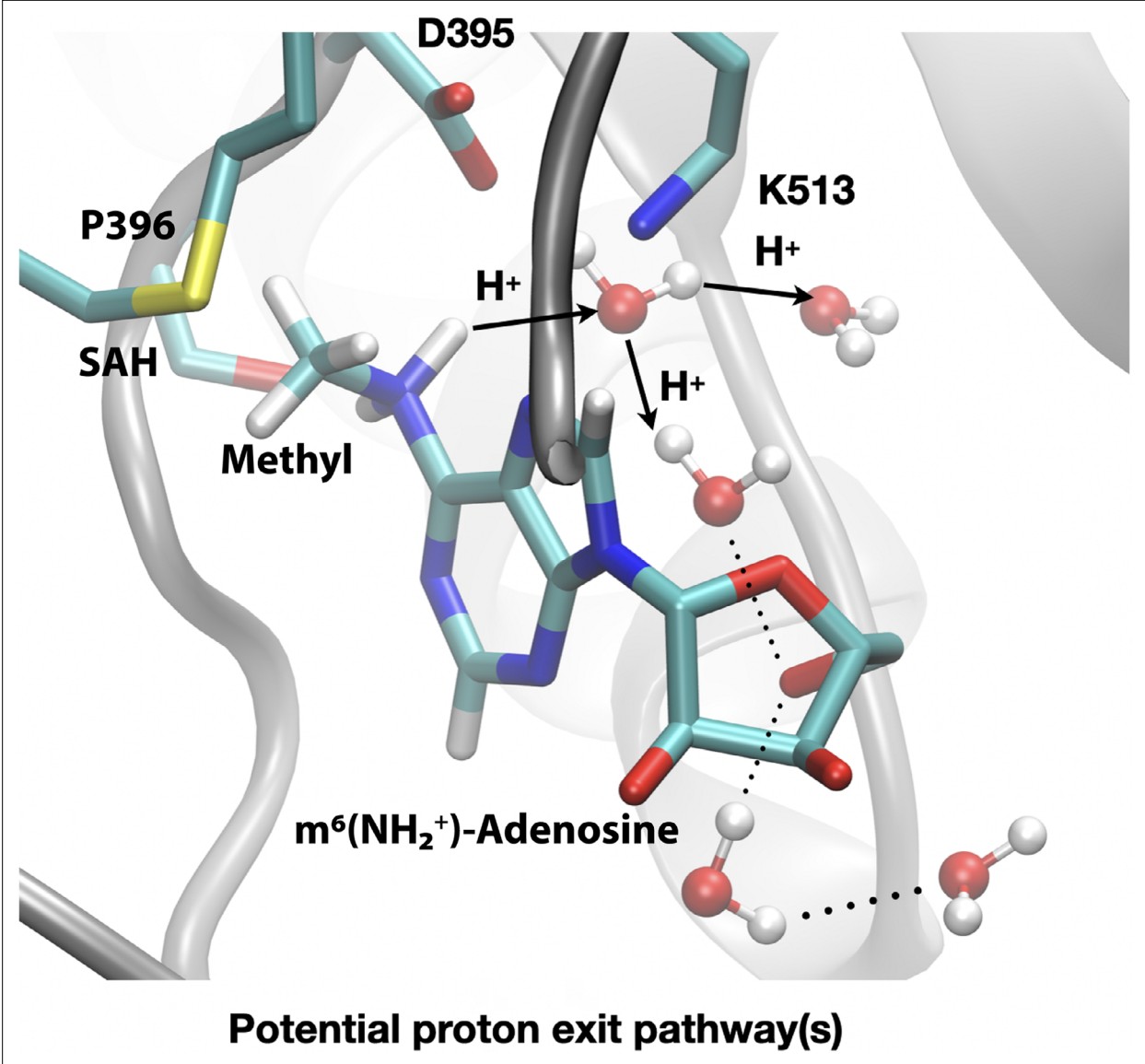

**Figure 9.** Deprotonation of adenosine-m⁶NH₂⁺ may occur readily through water wires that connect the adenosine-N⁶ position to the protein-solvent interface. Shown is a snapshot of the product state from third-order density functional tight binding (DFTB3)/MM simulations illustrating that the deprotonation of adenosine-N⁶ following the methyl transfer may proceed along multiple water-mediated pathways that lead to the protein/solvent interface. Methyltransferase-like protein 3 (METTL3) backbone is shown in grey ribbon representation with side chains shown as sticks and labelled, water molecules are shown as spheres. S-Adenosylhomocysteine (SAH) and m⁶(NH₂⁺)-adenosine are shown as sticks and labelled, together with the transferred methyl group. Hydrogen bonds are indicated with dotted lines. The movement of protons through water channels is indicated with arrows.

The difference in the methylation energetics of adenosine in different protonation states shown in *Table 3* is consistent with the p$K_a$ difference of adenosine before and after methylation. Thus, the large difference suggests that adenosine-N⁶ becomes much more acidic following methylation, which is consistent with literature estimates of the p$K_a$ values of N⁶-protonated adenosine derivatives in the range of –3 to –10 (*Kettani et al., 1997*). Close inspection of the active site structure in the product state of DFTB3/MM simulations reveals that the proton release may occur readily through water wires that connect the adenosine-N⁶ position to the protein-solvent interface (*Figure 9*). Therefore, favourable salt bridges in the active site (e.g. D395-K513) do not have to break to allow the proton release following methylation of the substrate.

Finally, we note that the recently solved crystal structures of METTL4, which belongs to a subclade of MT-A70 family members of MTases, showed an active site very similar to that of METTL3 (*Luo et al., 2022*). A nearby ion pair (D233-K364) is engaged in a hydrogen-bonding network involving both the

substrate and SAM. Therefore, we expect that the catalytic mechanism discussed here applies also to METTL4 and potentially other MT-A70 family members of MTases.

## Complete atomistic model of METTL3 binding site plasticity and methyl transfer mechanism

The complementarity of the methodologies and the congruence of the experimental data and simulation results allow us to construct a model of the METTL3 catalytic reaction (*Figure 10*). This model shows that in the apo state (*Figure 10*, State 0), the side chain of K513 is involved in intramolecular interactions that stabilise the protein. SAM binding displaces the K513 side chain and brings it in the right conformation where it can form a hydrogen bond to the $N^7$ of the adenosine substrate (*Figure 10*, State 1). Space for adenosine to bind is further conditioned by the conformational change of the H512 side chain which blocks the catalytic site in the apo state, but is drawn out of the adenosine binding site to interact with H538 in the SAM-bound state. The proper recognition of adenosine is conditioned by the interaction with the aromatic side chain of Y406 to which the adenosine substrate can bind and stabilise the ASL1 through interaction in the BA4 conformation (*Figure 10*, State 2). The Y406 side chain acts as gatekeeper and swaps out to allow the adenosine ring to flip and slip into the catalytic site where it is stabilised through hydrogen bonds to E481 and K513 in the BA2 conformation (*Figure 10*, State 3). The bond between K513 and adenosine is especially stable in the MD simulations started from the binding mode of BA2. Hydrogen bonds with the D395 side chain and P396 backbone enhance the nucleophilicity of the adenosine-$N^6$ and trigger the SN2 reaction with the electrophilic methyl group (*Figure 10*, State 4). The deprotonation of adenosine-$m^6NH_2^+$ following the methyl transfer may proceed along multiple water-mediated pathways that lead to the protein-solvent interface (*Figure 10*, State 5). The deprotonated $m^6A$ then most likely loses the hydrogen bond to the side chain of D395 as seen in the BA2 structure (*Figure 10*, State 6). The $m^6A$ product can then slip out and flip back into the BA4 conformation (*Figure 10*, State 7) before the flexibility of the ASL1 then facilitates its release, and SAH may also be released to yield again the apo state of METTL3 ready for the next catalytic cycle.

## Conclusions

We have conducted a combined experimental and computational study of METTL3-14 to characterise the methyl transfer reaction from SAM to the $N^6$ of adenosine in RNA. Crystal structures of METTL3-14 complexed with bisubstrate analogues (BAs) were solved and allowed us to carry out classical MD simulations and QM/MM free energy calculations. The encounter complex between the METTL3-SAM holo complex and RNA is promoted by electrostatic steering of the negatively charged RNA backbone by a surface groove of positively charged electrostatic potential centred around the sulphonium ion of SAM. The crystal structures together with mutational analysis and MD simulations have revealed a key role of METTL3 residue Y406 in recruiting the adenosine of the RNA substrate into a catalytically competent position and orientation. The proper recognition of adenosine is conditioned by interaction with the aromatic side chain of Y406 positioned in the flexible active site loop of METTL3. The side chain of Y406 recruits the RNA-adenosine, accommodates it into the catalytic site, and facilitates product release after methyl transfer. The adenosine ring forms hydrogen bonds to the side chains of E481 and K513 in the catalytic site of METTL3. Alanine mutants of these newly identified adenosine binding residues show abolished MTase activity compared to the WT METTL3-14. Importantly, these mutants are still folded and able to bind the SAH cofactor. This confirms the contribution of these newly identified adenosine binding residues to RNA-substrate binding, as shown also by the stability of their hydrogen bonds to adenosine in the MD simulations. Hydrogen bonds with the METTL3 D395 side chain and P396 backbone enhance the nucleophilicity of the adenosine-$N^6$ and trigger the SN2 reaction with the electrophilic methyl group of SAM. The QM/MM calculations provide evidence that the transfer of the methyl group from SAM to adenosine proceeds without prior deprotonation of the adenosine-$N^6$. Furthermore, the height of the QM/MM free energy barrier indicates that the methyl transfer step is not rate-determining. MD simulations suggest that the release of the coproduct SAH is the rate-limiting step.

In conclusion, the present study provides evidence that BAs and multiscale atomistic simulations can be used to decipher RNA recognition by human RNA MTases. The multidisciplinary strategy

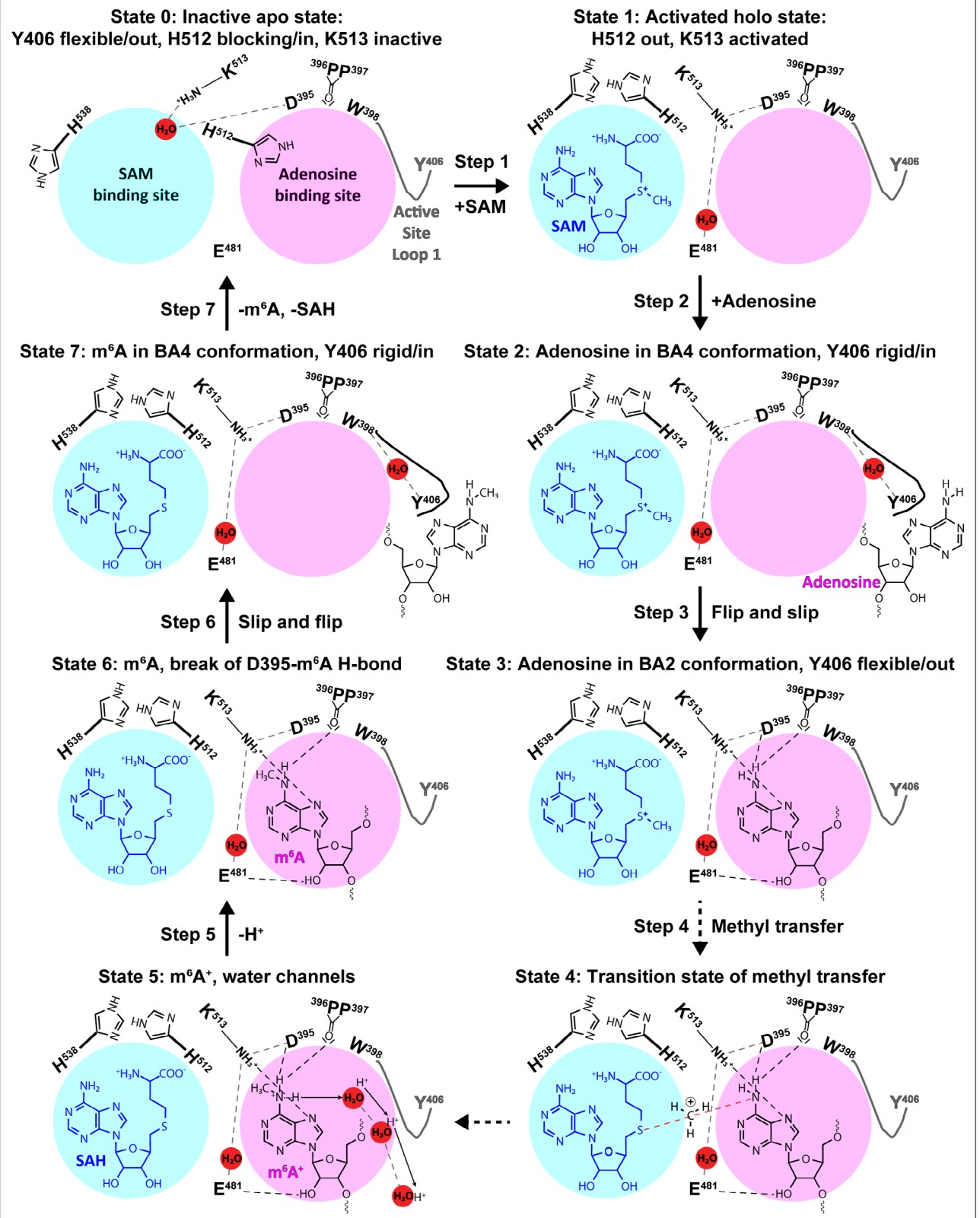

**Figure 10.** The experimental and computational data elucidate the individual steps of substrate binding, product release, and methyl transfer catalysed by methyltransferase-like protein 3 (METTL3). Schematic illustration of the individual steps making up the (co)substrate binding, methyl transfer, and (co)product release mechanism of METTL3; dashed grey and black lines indicate polar contacts that are intramolecular in METTL3 (including water mediated [waters shown as red spheres]) and intermolecular to the substrate/product adenosine/m6A, respectively. The ASL1 containing Y406 is either

*Figure 10 continued on next page*

*Figure 10 continued*

flexible (grey colour) or stabilised by the interaction with the substrate/product adenosine/m⁶A (black colour). Step 1: binding of *S*-adenosylmethionine (SAM) and flexibility of Y406 (supporting evidence from crystal structures and molecular dynamics [MD] simulations of apo and SAM-bound states); Step 2: substrate recognition (crystal structure of the complex with BA4); Step 3: flip of the adenosine ring and slip of the substrate into the catalytic site (crystal structure of the complex with BA2); Steps 4 and 5: methyl transfer and deprotonation of adenosine-m⁶NH$_2^+$ through water channels (movement of protons indicated by arrows) (quantum mechanics/molecular mechanics [QM/MM] free energy calculations); Steps 6 and 7: slip and flip of the product N⁶-methyladenosine (m⁶A) out of the catalytic site and subsequent dissociation from METTL3, along with *S*-adenosylhomocysteine (SAH) release to re-establish the apo state (MD simulations).

described here can be used for other RNA/DNA MTases to probe their active site by adapting the RNA/DNA sequence of the BA to the RNA/DNA substrate of the MTase of interest.

## Materials and methods

### Chemical synthesis of bisubstrate analogues (BAs)

The synthesis of the BAs was as previously described: BA1/2/3/4/6, *Atdjian et al., 2018*; Compound 12, *Atdjian et al., 2020*; GA*, *Meynier et al., 2022*.

### METTL3-14 expression, purification, and site-directed mutagenesis

For determining the half maximal inhibitory concentration (IC$_{50}$) with the full-length complex and for crystallisation studies with the truncated complex METTL3$^{MTD}$:METTL14$^{MTD}$ containing just the methyltransferase domains (MTD) of METTL3 (residues 354–580) and METTL14 (residues 107–395), the recombinant complex constructs were expressed using the baculovirus/Sf9 insect cell expression system and purified as described previously (*Śledź and Jinek, 2016*).

For mutational analysis, alanine mutants were generated in the baculovirus vector pFastBacDual-StrepII-GFP-TEV-METTL3-His-TEV-METTL14 using the QuikChange site-directed mutagenesis protocol and confirmed by sequencing. Recombinant baculoviruses were generated using the Bac-to-Bac system. For protein expression, suspension cultures of Sf9 cells in Sf-90 II SFM medium (Thermo Fisher) were infected at a density of 2×10⁶ ml⁻¹. Cells were harvested 72 hr post infection, resuspended in Buffer A (50 mM Tris-HCl pH 8.0, 500 mM NaCl) supplemented with Protease Inhibitor Cocktail (Roche Diagnostics GmbH, Germany), phenylmethylsulfonyl fluoride, Salt Active Nuclease (Merck), and lysed by sonication. The protein complex was purified by Ni-affinity chromatography on a 5 ml HisTrap HP column (Cytiva) equilibrated and washed with Buffer A and eluted with 250 mM imidazole. Proteins were further purified by Strep-tag purification using a 5 ml StrepTrap XT column (Cytiva) equilibrated and washed with buffer A and eluted with 50 mM biotin. The affinity tags were removed by digestion with TEV protease overnight at 4°C, followed by further purification by size exclusion chromatography using a Superdex 200 Increase 10/300 GL column (Cytiva) in 20 mM Tris-Cl, pH 8.0, and 200 mM KCl. The proteins were concentrated, flash-frozen in liquid nitrogen, and stored at –80°C until further use.

### Protein crystallisation

The SAH-bound holo protein crystals of METTL3$^{MTD}$:METTL14$^{MTD}$ were obtained as previously described (*Śledź and Jinek, 2016*). The BAs were dissolved in DMSO at concentrations of 100 mM. Complex structures were solved by soaking BAs into holo protein crystals and replacing the bound SAH in the METTL3 catalytic pocket. First, 1 µl of the BA dissolved in DMSO was left overnight to evaporate the solvent at room temperature (RT). The next day, 1 µl of mother liquor containing 30% PEG-3350 and 200 mM Mg-acetate was added on top of the dried compound stamp. One holo crystal was then transferred into the mother liquor over the target compound stamp. After 16 hr of incubation at 22°C, the crystals were harvested and flash-frozen in liquid nitrogen.

### Data collection and structure solution

Diffraction data were collected at the PXIII beamline at the Swiss Light Source (SLS) of the Paul Scherrer Institute (PSI, Villigen, Switzerland). Data were processed using XDS (*Kabsch, 2010*). The crystal structures were solved by molecular replacement by employing the 5L6D structure as the search model in the Phaser program (Phenix package) (*McCoy et al., 2007*). Crystallographic models

were constructed through iterative cycles of manual model building with COOT and refinement with Phenix.refine (*Emsley et al., 2010*; *Afonine et al., 2012*; *Liebschner et al., 2019*; *Emsley and Cowtan, 2004*).

## Reader-based TR-FRET assay

The inhibitory potencies of the BAs for METTL3 were quantified by a homogeneous time-resolved fluorescence (HTRF)-based enzyme assay as previously described (*Wiedmer et al., 2019*). Briefly, the level of m6A in an RNA substrate after the reaction catalysed by METTL3-14 was quantified by measuring specific binding to the m6A reader domain of YTHDC1 (residues 345–509) by HTRF. BAs that inhibit METTL3 decrease the m6A level and thus reduce the HTRF signal. Dose-response curves of titrations with the BAs were plotted in OriginLab 2018 and fitted with nonlinear regression 'log(inhibitor) vs. normalised response with variable slope' from which $IC_{50}$ values were determined.

For the mutational analysis, the HTRF assay was used with some modifications. In the reaction step, METTL3-14 (WT or mutant) (40 nM final concentration) methylates the 5'-biotinylated ssRNA (5'-AAGAACCGGACUAAGCU-3' (Microsynth)) (200 nM final concentration). The cosubstrate SAM (Cisbio, 62SAHZLD) (450 nM final concentration) was added as the last component and thus initiated the methylation reaction. The final reaction volume was 15 µl in 20 mM Tris-HCl, pH 7.5, 0.01% (wt/vol) bovine serum albumin (BSA). The reaction was let to incubate for 1 hr at RT and then stopped by addition of 5 µl detection buffer (50 mM HEPES, pH 7.5, 150 mM NaCl, 200 mM KF, 0.05% [wt/vol] BSA, 25 nM GST-tagged m6A reader YTHDC1(345–509), 3 nM XL665-conjugated streptavidin [Cisbio, 610SAXLB], 1× anti-GST $Eu^{3+}$-labelled antibody [from 400× stock, Cisbio, 61GSTKLB]). Capture of the m6A-modified RNA by the m6A reader and the biotinylated RNA by streptavidin was allowed to proceed for 3 hr at RT and in the dark before the HTRF signal was measured using a Tecan Spark plate reader (Tecan). The plate reader recorded with a delay of 100 µs the emission at 620 and 665 nm after the excitation of the HTRF donor with UV light at 320 nm. The emission signal was read over an integration time of 400 µs. For calculating ΔF ((($ratio_{sample}$ – $ratio_{background}$)/$ratio_{background}$) * 100), the reaction without SAM served as an internal control and as background signal.

## Thermal shift assay (TSA)

Experiments were conducted as previously described with some modifications (*Moroz-Omori et al., 2021*). Briefly, METTL3-14 (WT or mutants) at a final concentration of 0.5 µM was mixed with SAH at a final concentrations of 500 µM in a final volume of 20 µl in a buffer consisting of 20 mM Tris-Cl, pH 8.0, and 200 mM KCl. DMSO concentration was kept at 1% (vol/vol). SYPRO Orange was added at a final dilution of 1:1000 (vol/vol) as a fluorescence probe (ex/em 465/590 nm). TSA was performed on a LightCycler 480 Instrument II (Roche Diagnostics, Indianapolis, IN, USA). The temperature was raised in steps of 3.6°C per minute from 20°C to 85°C and fluorescence readings were taken in 0.1°C intervals. The $T_m$ values were determined as the transition midpoints of the individual samples. The $\Delta T_m$ values were calculated as the difference between the transition midpoints of the individual samples and the reference wells containing the protein and DMSO only from the same plate. Samples were measured in triplicates.

## MD simulations

The crystal structures of the METTL3-14 heterodimer in complex with the BA2 and BA4 ligands were used as starting conformation for simulating the heterodimer with its substrates SAM and AMP and products SAH and m6AMP. The addition of the phosphate to adenosine and m6-adenosine aims to mimic one element of the substrate RNA chain. The ligands were aligned to their respective moiety of the BA, keeping the original coordinates for present atoms, and reconstructing the missing parts. The missing segments of the METTL3-14 crystal structures were reconstructed using the SWISSMODEL web server with the structures as templates (*Waterhouse et al., 2018*). The simulated construct spans residues L354 to L580 of METTL3 and S104 to L289 of METTL14. All MD simulations were performed with GROMACS 2021.5 (*Abraham et al., 2015*) using the CHARMM36m July 2021 force field (*Huang et al., 2017*). The models were solvated in a 9 nm box and equilibrated with $Na^+$ and $Cl^-$ ions to a concentration of 150 mM. Energy minimisation was applied and a canonical equilibration under all-atom positional restraints was performed for 5 ns to reach 300 K. A further canonical equilibration was performed for 10 ns with the Cα atoms of the proteins, the adenine moiety of SAM/SAH and of AMP/

m⁶AMP under positional restraints. These partial restraints were set to allow a relaxation of the side chains around the ligands. Sixteen independent runs of canonical MD simulations were then started for each of the systems, sampling 500 ns per run.

The distance of the salt bridges observed in the crystals were monitored throughout the runs. Furthermore, the bound state was defined individually for the BA2 and the BA4 conformations. The distances between non-hydrogen atoms of protein and ligand were calculated and contacts were defined as intermolecular distances smaller than 5 Å. For each of the simulation groups, the binding pocket was determined as the residues with a mean contact presence higher than the 90 percentiles of mean contacts. A dissociation event was defined as the mean of the ligand to pocket distances surpassing a threshold of 10 Å. The dissociation rates were predicted using exponential fitting, modelled as a single exponential. A single exponential with a multiplicative factor was used to check the quality of the fitting, with a preexponential factor close to 1 indicative of a good fit. No fitting was done for SAM/SAH as they remained bound in almost every single trajectory. The rates were calculated by fitting the dissociation time of two blocks of the trajectories at a time. The 16 choose 8 (12,870) possible combinations were considered, and the mean of all the values was reported as the dissociation rate in *Table 2*. The standard deviation was used as error. The unbinding rates using all the trajectories without block averaging are shown in *Figure 4—figure supplement 2*.

## QM model calculations

To understand the intrinsic energetics of the methyl transfer reaction, we conducted QM calculations of an infinitely separated model substrate (adenosine) and a truncated model for the cofactor SAM in which the adenosine and amino moieties were replaced by ethyl groups. The $N^6$ position of the model substrate was taken to be either protonated ($-NH_2$) or deprotonated. Calculations were performed in the gas phase using two different density functional theory (DFT) methods (B3LYP with the D3 dispersion correction [*Becke, 1988*; *Becke, 1993*; *Lee et al., 1988*; *Grimme et al., 2010*] and $\omega$B97XD [*Chai and Head-Gordon, 2008*]) and two different basis sets (aug-cc-pVDZ and aug-cc-pVTZ) (*Dunning, 1989*; *Kendall et al., 1992*; *Woon and Dunning, 1993*). Calculations were also conducted at the DFTB3/3OB level for calibration as the same QM method is used in subsequent QM/MM free energy simulations. To probe the effect of solvation on the methyl transfer energetics, single point calculations were carried out with the conductor-like polarisable continuum model (*Barone and Cossi, 1998*; *Cossi et al., 2003*) using the gas-phase optimised structures at both B3LYP-D3 and $\omega$B97XD levels. DFT calculations using B3LYP-D3 and $\omega$B97XD were conducted using the Gaussian16 software (*Frisch et al., 2016*), and DFTB3 (*Gaus et al., 2011*) calculations were carried out using the CHARMM program (*Brooks et al., 2009*).

## QM/MM free energy simulations

We employed QM/MM free energy simulations to probe the mechanism of adenosine-$N^6$ methylation catalysed by the METTL3-14 complex. As illustrated in *Figure 3C*, the crystal structure (at 2.3 Å resolution) of the complex with a transition state analogue (the bisubstrate analogue BA2) suggests a rather straightforward mechanism through which the methyl group is transferred from SAM to the adenosine-$N^6$ position. The deprotonation of $N^6$ by a nearby base in principle may occur either before or after the methyl transfer, but QM/MM calculations strongly suggest that deprotonation occurs after the methyl transfer.

In the QM/MM simulations, the BA was first converted to a SAM non-covalently bonded to the adenosine: the nitrogen $N_{BS}$ atom in the crystal structure was replaced by a sulphur atom and the extra carbon atom was deleted. The O5*-PA bond was cut off and a hydrogen atom was patched to the location. The QM region included the cofactor SAM, the model substrate adenosine, the carbonyl groups of A394, D395, P396, and P397 without the backbone carbonyl group, and the side chain of K513. Link atoms were added between C and $C_\alpha$ of A394 and P397, and between $C_\beta$ and $C_\alpha$ of K513 to saturate the valence of the QM boundary atoms using the divided frontier charge (DIV) scheme (*König et al., 2005*). The QM atoms were treated with the DFTB3 method with the 3OB parameter set (*Gaus et al., 2013*; *Gaus et al., 2014*); benchmark calculations using model compounds (see *Table 3*) indicate that the DFTB3/3OB method describes the energetics of the methyl transfer reaction rather well in comparison to DFT calculations with a large basis set. The MM region was described with the CHARMM36 force field for proteins (*Huang and MacKerell, 2013*).

In the generalised solvent boundary potential (*Im et al., 2001*; *Schaefer et al., 2005*) framework, the inner region contained atoms within a 27 Å radius sphere centred at the $N^6$ in adenosine. Newtonian equations of motion were solved for atoms within 25 Å. Protein atoms in the buffer region (25–27 Å) were harmonically restrained with force constants determined from the crystallographic B factors and Langevin equations of motion were solved with a bath temperature of 300 K (*Brooks and Karplus, 1983*). The remaining portion of the system in the outer region was frozen. All water molecules were subject to a weak geometrical (GEO) type of restraining potential to keep them inside the inner sphere (*Brooks et al., 2009*). Weak GEO restraints were added on adenosine to make sure it was well bounded during the simulations. Electrostatic interactions among inner region atoms were treated with extended electrostatics and a group-based cut-off scheme (*Stote et al., 1991*). The static field due to the outer region atoms was evaluated with the linearised Poisson-Boltzmann (PB) equations using a focusing scheme, which employed a coarse grid of 1.2 Å and a fine grid of 0.4 Å (*Im et al., 1998*). The reaction field matrix was evaluated using spherical harmonics up to the 20th order. In the PB calculations, dielectric constants of the protein and water were set to 1 and 80, respectively, and the salt concentration was set to zero.

To probe the energetics of the methyl transfer reaction catalysed by the METTL3-14 complex, well-tempered multiple-walker metadynamics simulations (*Valsson et al., 2016*; *Barducci et al., 2008*) were carried out using the PLUMED-CHARMM interface (*Bonomi et al., 2009*; *PLUMED consortium, 2019*). The antisymmetric stretch that describes the methyl transfer process between the $N^6$ in adenosine and the SAM sulphur-methyl group was chosen as the collective variable (CV), i.e., $\xi = r(CE{-}SD) - r(CE{-}N6)$. The corresponding C-S distance and C-N distance are also monitored but no bias potential was added. SHAKE was applied to all bonds involving hydrogen and used to avoid undesired proton transfer reactions (*Ryckaert et al., 1977*).

The first two metadynamic runs were not well tempered for the efficiency of sampling. In the subsequent well-tempered runs, the bias factor was set to be 35. A new Gaussian biasing potential was added every 0.2 ps with an initial height of 0.3 kJ/mol and a width of 0.05. Twenty-four walkers with different initial velocities were used per simulation in parallel while sharing hill history among all walkers every 1 ps. Each walker was run for 250 ps for a total of 6 ns of sampling, and convergence was evaluated by comparing the PMF (potential of mean force) as a function of the number of Gaussians added.

## Acknowledgements

This work was supported by a grant of the Swiss National Science Foundation to AC (grant number 310030-212195). The MD simulations were carried out at Eiger@Alps at the Swiss National Supercomputing Center (in Lugano, Switzerland). This work was further supported by a grant of the French Agence Nationale de la Recherche (ANR), Project ARNTools, to MEQ (grant ARNtools-19-CE07-0028-01). The QM/MM study was supported by the NIH Grant R35-GM141930 to QC. Computational resources from the project BIO230101 allocated through ACCESS are greatly appreciated; part of the computational work was performed on the Shared Computing Cluster which is administered by Boston University's Research Computing Services (https://www.bu.edu/tech/support/research/).

## Additional information

### Competing interests

Qiang Cui: Senior editor, *eLife*. The other authors declare that no competing interests exist.

### Funding

| Funder | Grant reference number | Author |
| --- | --- | --- |
| Swiss National Science Foundation | 310030−212195 | Amedeo Caflisch |
| Agence Nationale de la Recherche | ARNtools-19-CE07-0028-01 | Mélanie Ethève-Quelquejeu |

| Funder | Grant reference number | Author |
|--------|------------------------|--------|
| National Institutes of Health | R35-GM141930 | Qiang Cui |

The funders had no role in study design, data collection and interpretation, or the decision to submit the work for publication.

### Author contributions

Ivan Corbeski, Pablo Andrés Vargas-Rosales, Jiahua Deng, Formal analysis, Validation, Investigation, Visualization, Writing – original draft, Writing – review and editing; Rajiv Kumar Bedi, Yaozong Li, Danzhi Huang, Investigation; Dylan Coelho, Emmanuelle Braud, Laura Iannazzo, Validation, Investigation; Mélanie Ethève-Quelquejeu, Resources, Supervision, Funding acquisition, Validation, Methodology; Qiang Cui, Formal analysis, Supervision, Funding acquisition, Validation, Investigation, Visualization, Methodology, Writing – original draft, Writing – review and editing; Amedeo Caflisch, Conceptualization, Resources, Supervision, Funding acquisition, Methodology, Writing – original draft, Writing – review and editing

### Author ORCIDs

Ivan Corbeski ⓘ http://orcid.org/0000-0002-5881-8425
Jiahua Deng ⓘ http://orcid.org/0000-0001-8865-4786
Yaozong Li ⓘ http://orcid.org/0000-0002-5796-2644
Mélanie Ethève-Quelquejeu ⓘ http://orcid.org/0000-0002-4105-3243
Qiang Cui ⓘ http://orcid.org/0000-0001-6214-5211
Amedeo Caflisch ⓘ https://orcid.org/0000-0002-2317-6792

Reviewer #1 (Public Review): https://doi.org/10.7554/eLife.92537.3.sa1
Reviewer #2 (Public Review): https://doi.org/10.7554/eLife.92537.3.sa2
Reviewer #3 (Public Review): https://doi.org/10.7554/eLife.92537.3.sa3
Author Response https://doi.org/10.7554/eLife.92537.3.sa4

# Additional files

### Supplementary files

• Supplementary file 1. Crystallography data collection and refinement statistics. Statistics for the highest-resolution shell are shown in parentheses.

• Transparent reporting form

### Data availability

All data needed to evaluate the conclusions in the paper are present in the paper and/or the supplementary materials. The coordinates of the METTL3-14-bisubstrate analogue complexes have been deposited in the Protein Data Bank under accession numbers 8PW9 (BA1 complex), 8PW8 (BA2 complex), 8PWA (BA4 complex), 8PWB (BA6 complex). The MD simulation trajectories and the PLUMED file that was used for the DFTB3/MM metadynamics simulations (plumed.dat) are available via Zenodo under the link: https://doi.org/10.5281/zenodo.10277884.

The following datasets were generated:

| Author(s) | Year | Dataset title | Dataset URL | Database and Identifier |
|-----------|------|---------------|-------------|-------------------------|
| Vargas Rosales PA | 2023 | Simulations of METTL3/METTL14 in complex with SAM+ADE or SAH+m6ADE | https://doi.org/10.5281/zenodo.10277884 | Zenodo, 10.5281/zenodo.10143263 |
| Bedi RK, Etheve-Quelquejeu M, Caflisch A | 2023 | Crystal structure of the human METTL3-METTL14 in complex with a bisubstrate analogue (BA1) | https://www.rcsb.org/structure/8PW9 | RCSB Protein Data Bank, 8PW9 |

*Continued on next page*

*Continued*

| Author(s) | Year | Dataset title | Dataset URL | Database and Identifier |
|---|---|---|---|---|
| Bedi RK, Etheve-Quelquejeu M, Caflisch A | 2023 | Crystal structure of the human METTL3-METTL14 in complex with a bisubstrate analogue (BA2) | https://www.rcsb.org/structure/8PW8 | RCSB Protein Data Bank, 8PW8 |
| Bedi RK, Etheve-Quelquejeu M, Caflisch A | 2023 | Crystal structure of the human METTL3-METTL14 in complex with a bisubstrate analogue (BA4) | https://www.rcsb.org/structure/8PWA | RCSB Protein Data Bank, 8PWA |
| Bedi RK, Etheve-Quelquejeu M, Caflisch A | 2023 | Crystal structure of the human METTL3-METTL14 in complex with a bisubstrate analogue (BA6) | https://www.rcsb.org/structure/8PWB | RCSB Protein Data Bank, 8PWB |

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
