## [Editor Report · eLife assessment]

This **important** study combines experimental and computational data to address crucial aspects of RNA methylation by a vital RNA methyltransferase (MTase). The authors have provided **compelling**, strong evidence, utilizing well-established techniques, to elucidate aspects of the methyl transfer mechanism of methyltransferase-like protein 3 (METTL3), which is a part of the METTL3-14 complex. This work will be of broad interest to biochemists, biophysicists, and cell biologists alike.

---

## [Referee Report · Reviewer #1 (Public Review)]

Summary:

This important study nicely integrates a breadth of experimental and computational data to address fundamental aspects of RNA methylation by an important for biology and health RNA methyltransferases (MTases).

Strengths: The authors offer compelling and strong evidence, based on carefully performed with appropriate and well-established techniques to shed light on aspects of the methyl transfer mechanism of the methyltransferase-like protein 3 (METTL3), which is part of the methyltransferase-like proteins 3 & 14 (METTL3-14) complex.

There are no weaknesses that we identified in the revised version.

---

## [Referee Report · Reviewer #2 (Public Review)]

Summary:

Caflisch and coworkers investigate the methyltransferase activity of the complex of methyltransferase-like proteins 3 and 14 (METTL3-14). To obtain an high resolution description of the complete catalytic cycle they have carefully designed a combination of experiments and simulations. Starting from the identification of bisubstrate analogues (BAs) as binder to stabilise a putative transition state of the reaction they have determined multiple crystal structures and validated relevant interactions by mutagenesis and enzymatic assays.

Using the resolved structure and classical MD simulations they obtained a kinetic picture of the binding and release of the substrates. Of note, they accumulate very good statistics on these processes using 16 simulation replicates over a time scale of 500 ns. To compare the time scale of the release of the products with that of the catalytic step they performed state-of-the-art QM/MM free energy calculations (testing multiple levels of theory) and obtain a free energy barrier that indicates how the release of the product is slower than the catalytic step.

Strengths:

All the work proceeds through clear hypothesis testing based on a combination of literature and new results. Eventually, this allows them to present in Figure 10 a detailed step-by-step description of the catalytic cycle. The work is very well crafted and executed.

---

## [Referee Report · Reviewer #3 (Public Review)]

Summary:

The manuscript by Coberski et al describes a combined experimental and computational study aimed to shed light on the catalytic mechanism in a methyltransferase that transfers a methyl group from S-adenosylmethionine (SAM) to a substrate adenosine to form N6-methyladenosine (m6A).

Strengths:

The authors determine crystal structures in complex with so-called bi-substrate analogs that can bridge across the SAM and adenosine binding sites and mimic a transition state or intermediate of the methyl-transfer reaction. The crystal structures suggest dynamical motions of the substrate(s) that are examined further using classical MD simulations. The authors then use QM/MM calculations to study the methyl-transfer process. Together with biochemical assays of ligand/substrate binding and enzyme turnover, the authors use this information to suggest what the key steps are in the catalytic cycle. The manuscript is in most places easy to read.

Weaknesses:

After revising the manuscript, there are few weaknesses beyond those listed in the paper.

---

## [Author Response]

The following is the authors’ response to the original reviews.

**Public Reviews:**

**Reviewer #1 (Public Review):**
Summary:This important study nicely integrates a breadth of experimental and computational data to address fundamental aspects of RNA methylation by an important for biology and health RNA methyltransferases (MTases).Strengths:The authors offer compelling and strong evidence, based on carefully performed work with appropriate and well-established techniques to shed light on aspects of the methyl transfer mechanism of the methyltransferase-like protein 3 (METTL3), which is part of the methyltransferase-like proteins 3 & 14 (METTL3-14) complex.Weaknesses:The significance of this foundational work is somewhat diminished mostly due to mostly efficient communication of certain aspects of this work. Parts of the manuscript are somewhat uneven and don't quite mesh well with one another. The manuscript could be enhanced by careful revision and significant textual and figure edits. Examples of recommended edits that would improve clarity and allow accessibility to a broader audience are highlighted in some detail below.

We thank the reviewer for the positive evaluation of our work. We have followed the suggestions and modified the text and figures as detailed further in our answers to the specific recommendations.

**Reviewer #2 (Public Review):**
Summary:Caflisch and coworkers investigate the methyltransferase activity of the complex of methyltransferaselike proteins 3 and 14 (METTL3-14). To obtain a high-resolution description of the complete catalytic cycle they have carefully designed a combination of experiments and simulations. Starting from the identification of bisubstrate analogues (BAs) as binders to stabilise a putative transition state of the reaction, they have determined multiple crystal structures and validated relevant interactions by mutagenesis and enzymatic assays.Using the resolved structure and classical MD simulations they obtained a kinetic picture of the binding and release of the substrates. Of note, they accumulated very good statistics on these processes using 16 simulation replicates over a time scale of 500 ns. To compare the time scale of the release of the products with that of the catalytic step they performed state-of-the-art QM/MM free energy calculations (testing multiple levels of theory) and obtained a free energy barrier that indicates how the release of the product is slower than the catalytic step.Strengths:All the work proceeds through clear hypothesis testing based on a combination of literature and new results. Eventually, this allows them to present in Figure 10 a detailed step-by-step description of the catalytic cycle. The work is very well crafted and executed.

We thank the reviewer for the positive evaluation of our work.

Weaknesses:To fulfill its potential of guiding similar studies for other systems as well as to allow researchers to dig into their vast work, the authors should share the results of their simulations (trajectories, key structures, input files, protocols, and analysis) using repositories like Zenodo, the plumed-nest, figshare or alike.

The reviewer is right. We have uploaded the simulation materials to Zenodo: the MD simulation data (trajectories, pdb files, parameter files), and the PLUMED file that was used for the DFTB3/MM metadynamics simulations. We provide the link in the “Data availability” section.

**Reviewer #3 (Public Review):**
Summary:The manuscript by Coberski et al describes a combined experimental and computational study aimed to shed light on the catalytic mechanism in a methyltransferase that transfers a methyl group from Sadenosylmethionine (SAM) to a substrate adenosine to form N6-methyladenosine (m6A).Strengths:The authors determine crystal structures in complex with so-called bi-substrate analogs that can bridge across the SAM and adenosine binding sites and mimic a transition state or intermediate of the methyltransfer reaction. The crystal structures suggest dynamical motions of the substrate(s) that are examined further using classical MD simulations. The authors then use QM/MM calculations to study the methyl-transfer process. Together with biochemical assays of ligand/substrate binding and enzyme turnover, the authors use this information to suggest what the key steps are in the catalytic cycle. The manuscript is in most places easy to read.

We thank the reviewer for the positive evaluation of our work.

Weaknesses:My main suggestion for the authors is that they show better how their conclusions are supported by the data. This includes how the electron density maps for example support the key interactions and water molecules in the active site and a better error analysis of the computational analyses.

We thank the reviewer for the comments and suggestions. We have followed the suggestions and added error analysis of the computational results as well as additional figures (in the supplementary information) that illustrate key interactions and water molecules in the active site supported by the electron density.

**Reviewer #1 (Recommendations For The Authors):**
The phrasing of the second sentence in the introduction is difficult to read. I am not sure it is necessary to define the DRACH motif if you are also giving the exact consensus sequence unless providing more context for other instances of the DRACH motif. Referring to this motif instead as "consensus sequence GGACU? may be more effective.

The reviewer is right. We corrected the sentence accordingly.

In the second paragraph of the introduction, a further short description of how METTL3-14 is "involved" in diseases would be appreciated.

We thank the reviewer for the comment. We made that clearer by including “by promoting the translation of genes involved in cell growth, differentiation, and apoptosis” together with a reference.

Is there any evidence that inhibiting METTL3-14 doesn't negatively impact healthy cells?

We thank the reviewer for the question. Yes, there is such evidence and we added to the sentence “but not in normal non-leukaemic haemopoietic cells” together with a reference to make this point clearer.

Bringing up the MACOM complex in the third paragraph of the introduction is perhaps not necessary unless further discussing the MACOM complex later.

The reviewer is right. We removed the mention of the MACOM complex.

Figure 1B: Color coding is difficult to distinguish on a screen and print out. More contrasting colors would be helpful.

We thank the reviewer for the suggestion. We removed the transparency from the protein cartoon representation that was the reason for the low contrast.

The level of detail in the "MD simulations for mechanistic studies of RNA MTases" is not advised. Would strongly encourage condensing this section to improve clarity and accessibility to a larger audience.

The reviewer is right. We removed non-essential parts of this paragraph.

Confirming the role of the hydroxyl in Y406 would be better supported by a Y406 -> F406 mutant because the A406 mutant could bind differently due to a loss of pi-stacking interactions.

The purpose of the Y406A mutant was to eliminate the interaction of the aromatic sidechain with adenosine as seen from the structure with BA4. Since there is no involvement of the Y406-OH group with adenosine, mutating to F did not seem sufficient. Furthermore, by mutating Y406 to alanine, we also eliminate the possibility for a water-mediated hydrogen bond to the W398 backbone. Hence, with the alanine mutant we achieve the strongest possible effect on the enzymatic activity while the integrity of the active site is maintained as seen from the thermal shift assay.

For Figure 4D, can the authors justify why SAH was used as a metric for SAM binding instead of using SAM directly? Additionally, referring to the RNA as "ligand" instead of "RNA" in the Figure caption is more confusing than simply calling it RNA.

We thank the reviewer for the comment. With the TSA, we wanted to show that with the adenosine binding mutants, the integrity of the METTL3 active site is still intact. It was shown that SAH is bound with higher affinity than SAM by METTL3 (DOI: 10.1016/j.celrep.2019.02.100). Since the magnitude of the thermal shift depends also on the affinity, we chose the higher-affinity binder SAH.There is no RNA per se shown in this figure. “Ligands” in the figure caption (A) refers to the three bound molecules that are shown and mentioned in the previous sentence: SAM, BA2, and BA4. “Ligand” in the figure caption (D) refers to “SAH” that was used in the experiment described and mentioned just after, but is now removed.

Figure 5D is very difficult to interpret. Removing the ribbons representing Y406 movement may make it easier to see. Color coding the Supplementary Movie 1 to match would be also helpful.

The reviewer is right. We have changed the figure to make the different conformations of METTL3 and its Y406 sidechain clearer. However, we left the coloring of the different conformations as the colors are connected to different time points of the simulation. Following the suggestion of the reviewer we changed the coloring of SAM and AMP to match that of the supplementary movie.

Figure 10 is overwhelming as is. Removing the grey area around the binding sites and toning down the color of the substrate binding sites would help with visibility. The size of the chemical structures and illustrations is currently too small to easily be made out. A full page-sized figure may be beneficial for this figure.

We agree with the reviewer and have changed the figure to make each reaction step clearer and better recognizable.

Minor >edits

Change "Despite the growing knowledge on the diverse pathways" to "Despite growing knowledge of the diverse pathways involving METTL3-14".

We corrected the sentence.

Perhaps use "redundant active site" instead of "degenerate active site".

We changed the word as suggested.

Consider moving "The METTL3 MTase domain has the catalytically active SAM binding site and adopts a Rossmann fold that is characteristic of Class I SAM-dependent MTases" to before "METTL14 also has an MTase domain, however, with a degenerate active site of hitherto unknown function, and so-called RGG repeats at its C-terminus essential for RNA binding" to keep information about METTL3 together.

We shifted the part of the text as suggested.

"Molecular dynamics studies have mainly focused on protein and bacterial MTases"? Does this mean bacterial MTases that methylate proteins?

We thank the reviewer for the comment. This means bacterial MTases in general. The example that we mention is of a bacterial MTase that methylates a chemical precursor. We changed the sentence slightly to make that clearer.

In "Bisubstrate analogues bind in the METTL3 active site", please consider the following:Change "and to investigate" to "and investigated".Briefly describe the enzymatic assay in the main text.Either more clearly defining "least potent" or change to "have the highest IC50 values".

We made all the suggested changes to improve the description of the assay and its outcomes.

In Figure 3, remove some of the amino acid labels from panels A, C, and E for clarity, especially since panels B, D, and F more clearly demonstrate the interactions.

We removed amino acids that were not involved in polar contacts and adapted the figure caption accordingly.

In panels 3D, 3F, and 4B, the lightning bolts are too small to make out as lightning bolts. An asterisk or other symbol may be easier to distinguish.

We made the lightnings more than double the size to make them better recognizable.

In Figure 4C, no units are provided on the y-axis. Additionally, I do not believe the arrows indicating "Loss of activity" are necessary.

These are arbitrary units as it is a ratio which is explained in the materials and methods section. We removed the arrows following the suggestion of the reviewer.

While demonstrating mutants with no activity still retain SAM binding is suggestive of the mutant impacting RNA binding, this would still be better supported with RNA binding studies. Electrophoretic mobility shift assays would be sufficient if Tm studies are time-consuming. While these experiments could be informative, we also acknowledge that they may be outside the scope of this current report.

We thank the reviewer for suggesting these experiments and acknowledging that they would be outside of the scope of the current study. Such RNA binding experiments can turn out to be very time consuming, both in TSA and EMSA. The reason is mainly this: The RNA substrate must be chosen such that it binds sufficiently strong to the WT to cause an effect (thermal shift or electrophoretic mobility shift), but also to observe a clear difference in binding between WT and mutant proteins. Since many more residues of METTL3 and METTL14 contribite to RNA binding, the effects of individual mutants on affinity might be too small to be confidently detected in TSA or EMSA. In particular, we only identified the substrate adenosine binding residues, and mutating them and hence preventing adenosine binding alone, might not have a big effect on overall RNA binding affinity. The enzymatic assay that we used, on the other hand, is more sensitive since the detection is fluorescence based and quantifies the conversion of A to m6A in an RNA substrate, and more factors than just affinity play a role for enzymatic activity, such as correct orientation and stability of the adenosine in the active site and stabilization of the transision state.

A written narrative to accompany Supplementary Movie 1 would make it much more accessible to those unfamiliar with modeling and simulations.

We thank the reviewer for the comment. We expanded the caption to the movie with a narrative describing different events at different time points in the movie.

Table 3 could be made clearer to those without MD experience by defining/indicating the top row as different computational models.

The reviewer is right. We have added a footnote to Table 3 to clearly indicate the different density functional theory and semi-empirical density functional tight binding method used in this study. We also added another line in the table.

In the conclusion, the authors state "the height of the QM/MM free-energy barrier indicates that the methyl transfer step is not rate-determining." How does this compare to experimental data? Additional kinetic assays to demonstrate this experimentally would go a long way in convincing the reader of this conclusion.

We thank the reviewer for the question. Kinetic assays have been performed for METTL3-14 and we mention and reference them in the text. We believe that further kinetic experiments would be outside of the scope of this study. Furthermore, the METTL3 mutants that we made show no activity in our enzymatic assay and hence kinetic studies would be probably impossible to do with them.

As we show from QM/MM and describe in the text, the methyl cation in the SAM cofactor is transferred directly to the N6 position of the adenosine substrate. DFTB3/MM free energy simulations show that this mechanism has an energetic barrier of 15-16 kcal/mol. The turnover as published based on an enzymatic assay is 0.2-0.6 min-1 at ambient temperature which implies a barrier of ~20 kcal/mol. This value is higher than that determined for the methyl transfer alone as determined by QM/MM. Hence, in the overall mechanism, there must be a step that is slower than the methy transfer and hence we conclude that the methyl transfer is not the rate-limiting step.

**Reviewer #3 (Recommendations For The Authors):**
I only have a few comments about the work.(1) It would be good if the authors could show more of the data that is used as the basis for their conclusions. For example, IC50 values are presented (Table 1) without error estimates or an indication of the quality of the data that is used to estimate the data.

We thank the reviewer for the suggestion. We included errors of the IC50 values and show the dose response curved from the enzymatic assay with the BAs as inhibitors in a new Supplementary Figure S1.

(2) More substantially, it would be good to have a more detailed analysis of the crystal structures in terms of the properties that are mentioned/analysed. While the structures are relatively good (2.1 Å2.5Å), it is not clear to the reader how this data supports the interactions that are proposed. For example, the authors pinpoint a number of hydrogen bonding interactions and water molecules in the complexes. They might consider showing support for some of these in the electron density maps. Similarly, it would be good to show the densities that support the substantial differences of the Ade in the BA2 and BA4 complexes. These might be supplementary files. I note also that the structures are not yet released or available for analysis [which of course is a valid choice but also means that I cannot inspect the maps myself].

We have added supplementary figures supporting the conformations of the BAs and their interactions with METTL3 with electron density, for BA1 and BA6 in Supplementary Figure S2, and for BA2 and BA4 in a new Supplementary Figure S3.

(3) It would be useful with an error analysis of the off-rates estimated from the MD simulations and a discussion of the accuracy of these estimates. Even the slower dissociation events seem quite fast. What are the rough affinities of these molecules and how fast would the binding need to be to be compatible with the affinity and estimated off-rates?

We expanded upon this in the results paragraph concerning the MD simulations. The affinities of METTL3-14 binding to AMP or m6AMP can be expected to be very low, with Kd values in the millimolar range. We have not measured these Kd values, nor have we found any published data, but we have conducted thermal shift assays with A and m6A and did not observe any significant thermal shifts in the melting temperature of METTL3-14 at high micromolar concentrations of these compounds, indicative of a very low binding affinity. This is to be expected because METTL3-14 should not methylate adenosines unspecifically but rather in the GGACU motif of substrate mRNA.

(4) The authors use QM/MM simulations with metadynamics to estimate the energy profile of the methyl transfer reaction. They find a barrier of ca. 15 kcal/mol and suggest this to be compatible with the enzymatic turnover rate of ca. 0.3/min. Here it would be good with a clearer description of the possible sources of error and assumptions in making these statements. First, what is the error on the estimated energy profile from the metadynamics? The authors mention the analysis of progression of the PMF as a function of time, but that is in itself not a strong test for convergence (the PMF may stay constant if there is little sampling). What does the time series of the CV look like? Second, it seems as if the authors are assuming a large pre-exponential factor (10^9/s ?). Is that correct, and how sure are they of this value? Finally, when linking the barrier of the methyl-transfer reaction to the overall turnover rate it sounds like they assume that other parts of the reaction do not affect the turnover rate. Is that correctly understood, and what is the evidence for that? It sounds like the authors are saying that step 5 in the cycle (Figure 10) is limiting.

We thank the reviewer for the questions. Accordingly, we have carried out additional simulations and statistical error analyses.

(i) We have carried out two additional sets of multi-walker metadynamics simulations with the same setup as the original calculation, except for using different initial random seeds. Using the three independent sets of metadynamics simulations, we can better estimate the statistical uncertainty for the computed potential of mean force (PMF). We have updated the PMF in Fig. 8b, in which the solid curve represents the result averaged over three independent runs, and the shaded area represents the standard error of the mean of the three replicas. The figure caption of Fig. 8b is revised accordingly.

(ii) To further illustrate the convergence behavior of the metadynamics simulations, we have included the following supplementary files: (1). Potentials of mean force computed with different numbers of deposited Gaussians are compared. (2). As suggested by the reviewer, we show the time series of the collective variable (CV) sampled by the 24 independent walkers during one set of metadynamics simulations. These results clearly indicate that the CV exhibits diffusive behaviors between the reactant and product regions, further supporting the adequate sampling and convergence of our metadynamics simulations.

(iii) Regarding the issue of pre-factor used in the rate estimate, we have indeed used the common approximation of kT/h as in the regular transition state theory. Many studies in the literature support the use of this expression for very localized chemical reactions in enzymes. We have included several representative references along this line: (1) M. Garcia-Viloca, J. Gao, M. Karplus, D. G. Truhlar, How enzymes work: Analysis by modern rate theory and computer simulations, Science, 303, 186-195 (2004) (2) D. R. Glowacki, J. N. Harvey, A. J. Mulholland, Taking Ockham’s razor to enzyme dynamics and catalysis, Nat. Chem. 4, 169-176 (2012)

(iv) Regarding the nature of the rate-limiting event, please see our response to reviewer 1.

(5) The authors should ideally make the input files for their simulations available and deposit the plumed files in for example plumed-nest (as indicated in their reference 100).

We agree with the reviewer. Accordingly, we have uploaded the PLUMED file that we have used for the DFTB3/MM metadynamics simulations (plumed.dat) together with the MD simulation trajectories to Zenodo.

Minor(1) Many of the details in Figure 10 are very small and difficult to read without zooming in. Consider whether some parts could be made larger.

The reviewer is right. We have changed the figure to make each reaction step clearer and better recognizable.